# The Effects of Multistage Fuel-Oxidation Chemistry, Soot Radiation, and Real Gas Properties on the Operation Process of Compression Ignition Engines

**Valentin Y. Basevich** [1], **Sergey M. Frolov** [1,2,*], **Vladislav S. Ivanov** [1,2], **Fedor S. Frolov** [1,2] and **Ilya V. Semenov** [2]

1   Department of Combustion and Explosion, Semenov Federal Research Center for Chemical Physics of the Russian Academy of Sciences, Moscow 119991, Russia; basevichv@yandex.ru (V.Y.B.); ivanov.vls@gmail.com (V.S.I.); f.frolov@chph.ru (F.S.F.)

2   Department of Computational Mathematics, Federal Science Center "Scientific Research Institute for System Analysis of the Russian Academy of Sciences", Moscow 117218, Russia; ilyasemv@yandex.ru

\*   Correspondence: smfrol@chph.ras.ru

**Abstract:** The objectives of the study are to reveal the influence of multistage fuel-oxidation chemistry, thermal radiation of soot during the combustion of a small (submillimeter size) fuel droplet, and real gas effects on the operation process of compression ignition engines. The use of the multistage oxidation chemistry of iso-octane in the zero-dimensional approximation reveals the appearance of different combinations of cool, blue, and hot flames at different compression ratios and provides a kinetic interpretation of these phenomena that affect the heat release function. Cool flames are caused by the decomposition of alkyl hydroperoxide, during which a very reactive radical, OH, is formed. Blue flames are caused by the decomposition of $H_2O_2$ with the formation of OH. Hot flames are caused by the chain branching reaction between atomic hydrogen and molecular oxygen with the formation of OH and O. So-called "double" cool flames correspond to the sequential appearance of a separated cool flame and a low-intensity blue flame rather than two successive cool flames. The use of a one-dimensional model of fuel droplet heating, evaporation, autoignition, and combustion at temperatures and pressures relevant to compression ignition engines shows that the thermal radiation of soot during the combustion of small (submillimeter size) droplets is insignificant and can be neglected. The use of real gas caloric and thermal equations of state of the matter in a three-dimensional simulation of the operation process in a diesel engine demonstrates the significant effect of real gas properties on the engine pressure diagram and on the NO and soot emissions: real gas effects reduce the maximum pressure and mass-averaged temperature in the combustion chamber by about 6 and 9%, respectively, increases the autoignition delay time by a 1.6 crank angle degree, increase the maximum heat release rate by 20%, and reduce the yields of NO and soot by a factor of 2 and 4, respectively.

**Keywords:** detailed kinetic mechanism; autoignition; cool and blue flames; compression ignition engine; operation process; numerical simulation; real gas equation of state; nitrogen oxides; soot

## 1. Introduction

In view of increasingly strict regulations on pollutant and greenhouse gas emissions [1], CFD methods are becoming increasingly important for simulating in-cylinder processes and for developing combustion strategies in transportation engines [2,3]. Predictability can be reached with adequate models of all accompanying phenomena, including gas exchange, mixture formation, combustion and pollutant formation, heat transfer to the walls, etc. In this manuscript, we focus on only studying the effects of the detailed chemistry of fuel oxidation, soot radiation during autoignition and combustion of small-size droplets in engine conditions, and real gas thermodynamics on the operation process of the compression ignition engine (CIE). Despite the fact that commercially available CFD codes provide

a possibility of applying the state-of-the-art detailed kinetic mechanisms (DKMs) of fuel oxidation, sophisticated soot thermal radiation models, as well as real gas equations of state (EoS), automotive engineers worldwide still apply either semi-empirical models or an overall reaction mechanism for simulating spray autoignition and combustion, zonal methods coupled with the oversimplified models of soot formation for handling radiative heat transfer, and the ideal gas EoS. However, both semi-empirical models and overall reaction mechanisms are not universal and are only applicable within the parametric domains they are developed for. Such models and mechanisms do not usually include low-temperature chemical transformations in the combustible mixture, which can have a noticeable effect on the rate of energy release and the engine operation process as a whole.

This paper has three objectives. The first is to give a kinetic interpretation of the multistage autoignition phenomena observed in CIEs. The second is to reveal the effect of soot thermal radiation on the autoignition and combustion of small (submillimeter size) fuel droplets under the conditions of CIEs. The third is to reveal the real gas effects on the operation process of CIEs and the yields of pollutants (soot and NO) using the accurate thermal and caloric EoS of the main species. These objectives and the obtained results are the novel and distinctive features of this paper. The next three sections in the Introduction provide selective literature reviews on these topics.

## 1.1. Multistage Autoignition

The concept of the multistage autoignition of a hydrocarbon fuel with separated "cool", "blue", and "hot" flames was introduced in [4] based on studies of the autoignition of iso-octane and other hydrocarbons under engine conditions. Under these conditions, at the parametric plane "mixture composition–compression ratio" (Figure 1), in addition to the domain of mixture autoignition, three more domains of pre-flame luminosity are identified. In the first, a partial reaction of the mixture is found to occur and is accompanied by an increase in pressure and temperature, a decrease in oxygen and fuel concentrations, and the formation of intermediate and final reaction products in certain amounts. The luminous flame of bluish color corresponding to this parametric domain is called a "blue" flame and is caused by the luminosity of excited formyl HCO∗. The second and third domains are located below the domain of blue flames at lower compression ratios. In these domains, a partial reaction is also found to proceed with some increase in the pressure and luminosity of excited formaldehyde $H_2CO∗$. The flames corresponding to these two domains are called a "double" cool flame and a single cool flame, respectively. In [5], based on the review of a large experimental material, a generalization was made and the concept of multistage autoignition with individual cool, blue, and hot flames was introduced. Modeling the kinetics of the autoignition of hydrocarbons under laboratory conditions using a DKM indicates that cool flames are caused by the decomposition of alkyl hydroperoxide, during which a very reactive radical, OH, is formed, and blue flames are caused by the decomposition of $H_2O_2$ with the formation of OH [6].

Recently, the DKMs of the oxidation of heavy hydrocarbons up to $C_{20}$ containing thousands of species and tens of thousands of elementary reactions have been proposed. For example, the authors of [7] proposed such a DKM composed of 7182 species and 31,721 reactions by merging two mechanisms reported in [8,9]. Later, this mechanism was reduced to contain 2277 reactions and 522 species [10]. Many of the proposed DKMs are capable of describing the low-temperature oxidation of hydrocarbons and predicting the formation of both cool and blue flames. The blue flames are often referred to as "intermediate temperature heat release" [11–15] or "intermediate temperature ignition" [16,17].

Despite the undoubted advantages of the DKMs, their application in multidimensional problems of combustion in engines is difficult because of their complexity. Moreover, consideration of all possible reactions between all possible species, including their isomers, would make such mechanisms much larger. For example, only the inclusion of vibrationally excited molecules with the reactions of their formation and consumption leads to multiple increases in the DKM size [18,19]. In addition, such DKMs involve many uncertainties

caused by the lack of reliable data on the reaction rate constants and thermodynamic functions. All these factors greatly affect the accuracy and suitability of the DKMs.

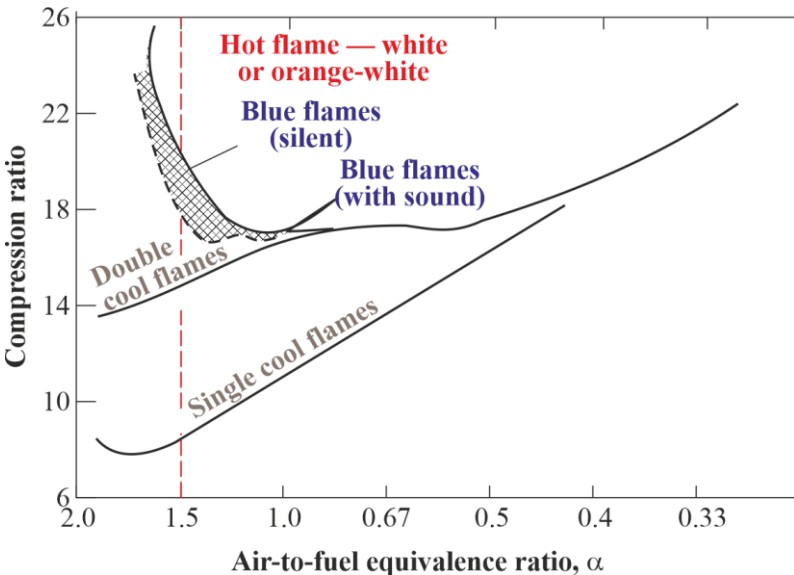

**Figure 1.** The boundaries of the single and "double" cool flames, blue flame, and hot flame domains during compression induced the autoignition of iso-octane–air mixtures in a compression ignition engine [4] (legends are taken from [5]). Initial temperature $T_0 = 343$ K, initial pressure $P_0 = 1$ bar, engine rotation speed $n = 1500$ rpm. The red dashed vertical line at $\alpha = 1.5$ corresponds to the composition of the fuel–air mixture examined in Section 3.1.

Thus, the known DKMs are generally imperfect and limited to a certain degree. However, in many cases, the optimal and compact DKMs that provide the overall reaction rate and the composition of key intermediate and final products with an acceptable accuracy are required rather than the maximal DKMs, including species and reactions unnecessary for problem solution. Such mechanisms still have the status of non-empirical DKMs, as all included elementary reactions are kinetically substantiated. When simulating the oxidation of hydrocarbon fuels, it is always possible to construct a DKM with a limited number of species and elementary reactions while retaining the main reaction pathways. Such a DKM is proposed for iso-octane [20], which describes the kinetics of multistage autoignition. This DKM has been used to compare the results of calculations with experimental data on the autoignition delay time, laminar flame structure and propagation velocity, and fuel droplet combustion in wide ranges of temperature, pressure, and mixture composition.

*1.2. Soot Formation and Radiation*

The radiative heat transfer is mainly caused by soot particles formed in fuel-rich locations of the combustion chamber. Contrary to carbon dioxide and water vapor molecules, which radiate in a narrow spectral band, soot particles emit radiation in a wide continuous spectrum, thus producing appreciably more intense radiation than that from the triatomic molecules. Therefore, the key problem in this respect is to evaluate the rate of soot formation and oxidation.

There are many publications in the literature on soot formation, oxidation, and radiation in conditions relevant to CIEs. There exist several noteworthy articles dealing with gaseous diffusion flames [21–25] and hydrocarbon droplet flames [26–28]. These articles discuss the existing approaches to describe the formation and growth of soot particle nuclei, their coagulation, activation and deactivation, and oxidation. As mentioned earlier in this paper, we focus herein on soot radiation in droplet flames under engine conditions. The video frames of large (several millimeters in size) droplet ignitions and combustions in microgravity indicate the appearance of a loose spherical "soot shell" between the flame

and the droplet composed of very fine soot particles [29]. This soot shell thickens over time due to the accumulation of soot particles and becomes a kind of thermal screen due to the extraction of a part of the heat flux from the flame to the droplet and thermal radiation to the ambiance. As a result, the rates of droplet evaporation and combustion decrease, and the flame quenches. This phenomenon is called "radiative flame quenching" [30] and has been experimentally [31] and theoretically [32] studied. Interestingly, after flame quenching, the droplet continues unexpectedly and quickly evaporates. Moreover, after some time, a brightly glowing flame spontaneously appears around the droplet and disappears again. Such flame flashes and extinctions can repeatedly occur during the droplet's lifetime. The phenomenology of radiative flame quenching with subsequent flashes of cool and blue flames was computationally reproduced in [33,34] for large n-heptane and n-dodecane droplets. The possibility of the radiative quenching of small (submillimeter-size) droplets is still questionable. Under terrestrial conditions, soot is entrained by the flow of hot gases caused by natural convection, and the flame-quenching ability of the soot shell is less pronounced.

*1.3. Real Gas Effects*

The ideal gas approximation is only applicable for gases with low density when the interaction between molecules is negligible. Thermodynamic conditions in modern turbocharged diesel engines can go beyond the limits of this approximation due to high gas pressure (up to 200 bar [35] or 140 bar [36]) or density (up to 200 kg/m$^3$ [37]). Localized regions with low temperatures and high pressure in an engine cylinder can be critical. For example, the maximum pressure in an engine cylinder can reach 200–300 bar, while the temperature in the vicinity of the cooled walls can be quite low (400–550 K). Under such conditions, the real gas effects may manifest themselves. Thus, real gas effects can significantly affect the behavior and structure of the flow in the region of diesel spray [37–40] and change ignition delay times in diesel engines, particularly in the region of the negative temperature coefficient [40,41].

In the literature, multiparameter EoS have usually been used to theoretically approximate the $p$–$\rho$–$T$ data with the accuracy of experimental data [42]. In handbooks, the $p$–$\rho$–$T$ EoS containing a few tens of parameters (up to 50) have been reported. Such EoS are nothing else but complex interpolations of experimental data. The application of such equations in CFD is accompanied by large computational costs, as the EoS is used at each iteration of each time step in each control volume. Such EoS have only been reported for lower n-alkanes.

There are many approximate real gas EoS, among which the modifications of the Van der Waals EoS are most popular; the Redlich–Kwong equation [43] and Peng–Robinson equation [44].

In modern diesel engines, effects other than real gas effects can also play a certain role. Among them are the two-phase equilibrium effects, the effects of the gas-phase chemical decomposition of fuel molecules, and the effects of dissociation and ionization. In the literature, there are no available approaches to take these effects into account in the real gas EoS. Therefore, these effects are commonly neglected.

The paper is organized as follows. In Section 2, we briefly introduce our own DKM of hydrocarbon fuel oxidation (Section 2.1), our own model of droplet autoignition and combustion (Section 2.1), and our own real gas EoS (Section 2.3) with examples of their validation. Thereafter, we briefly describe the procedures of the numerical solution of the new target problems, namely, the manifestation of multistage autoignition in the CIE (Section 2.4), the manifestation of soot thermal radiation during droplet combustion in CIE (Section 2.5), and the manifestation of real gas effects in CIE (Section 2.6). In Section 3, we show the results of calculations and discuss the corresponding implications. Section 4 summarizes the results of the study and we discuss the directions of future work.

## 2. Materials and Methods

### 2.1. Reaction Mechanism

The oxidation of hydrocarbons exhibits great generality [5]. The DKM of iso-octane oxidation and combustion was developed in [20,45]. This DKM includes the main processes that determine the reaction rate and the formation of the main intermediate and final reaction products. Since all included elementary reactions are kinetically substantiated, this DKM has the status of an ab initio mechanism. The DKM is based on the non-extensive approach and the analogy technique [46]. The key point of the non-extensive approach is the generality of the main reaction pathways. The analogy technique is applied for the selection of elementary reactions important for modeling the multistage oxidation. It is assumed, in the DKM, that the first addition of oxygen to the peroxy radical is sufficient, whereas the second addition of oxygen to the isomerized form of the peroxy radical is not essential for the overall reaction progress. The DKM of iso-octane oxidation is obtained by adding nine isomerized derivatives of iso-heptane (2,2-dimethylpentane) and iso-octane (2,2,4-tri-methylpentane) to each of the components of the DKM of oxidation of n-octane $C_8H_{18}$ and isomerized alkanes: iso-butane (2-methylpropane), iso-pentane (2-methylbutane), and iso-hexane (2-methylpentane). Their reactions with each other and other species available in the DKM are also added. The leading role is assumed to be played by deisomerization reactions to form stable intermediate isomerized molecules of 2,2-dimethylpentane, 2-methylpentane, 2-methylbutane, and 2-methylpropane and the corresponding normal-structure hydrocarbons, including methane. All of the other reactions leading to the increase in the "linear" five-membered portion of an iso-octane molecule are assumed to have no effect on the iso-octane oxidation rate. The resulting DKM of iso-octane oxidation is fairly compact: it includes 763 reactions involving normal-structure species and 987 reactions involving isomerized species. The total number of species involved in the DKM is 144.

The enthalpy $\Delta H^{\circ}_{f298}$, entropy $S^{\circ}_{298}(T)$, and the specific heat at constant pressure $C_{p0}(T)$ are calculated following known recommendations and additivity rules [47]. The Arrhenius parameters of rate constants for some reactions are calculated based on the rate constants for the reactions with normal-structure species due to the lack of the corresponding experimental data. For this purpose, a two-parameter form for the rate constant of an elementary reaction is used with the preexponential factor $A$ and activation energy $E$ given by [46]:

$$A_{i(i)} = A_{i(n)} \exp\left[\left(\Delta S_{i(i)} - \Delta S_{i(n)}\right)/R\right] \tag{1}$$

$$E_{i(i)} = E_{i(n)} - 0.25\left(\Delta H_{i(i)} - \Delta H_{i(n)}\right) \tag{2}$$

for exothermic reactions and

$$E_{i(i)} = E_{i(n)} + 0.75\left(\Delta H_{i(i)} - \Delta H_{i(n)}\right) \tag{3}$$

for endothermic reactions. Here, $\Delta S$ and $\Delta H$ are the entropy and enthalpy changes; indices $i$, $(i)$, and $(n)$ correspond to the reaction number, to the isomerized-structure species, and to the normal-structure species, respectively. Some corrections of the Arrhenius parameters thus obtained were needed for a limited number (less than ten) of reactions. The eminent feature of the DKM is the manifestation of the occurrence of both cool and blue flames at the low-temperature autoignition of iso-octane and other included hydrocarbons.

Figure 2 demonstrates the predictive capabilities of the DKM. It compares the calculated and measured ignition delays for the homogeneous stoichiometric iso-octane–air (Figure 2a) and n-heptane–air (Figure 2b) mixtures in wide ranges of initial conditions in terms of pressure and temperature. A comparison of the calculation results with the experimental data is seen to yield their satisfactory agreement, as the uncertainty in the ignition delay data can exceed 100% [48,49].

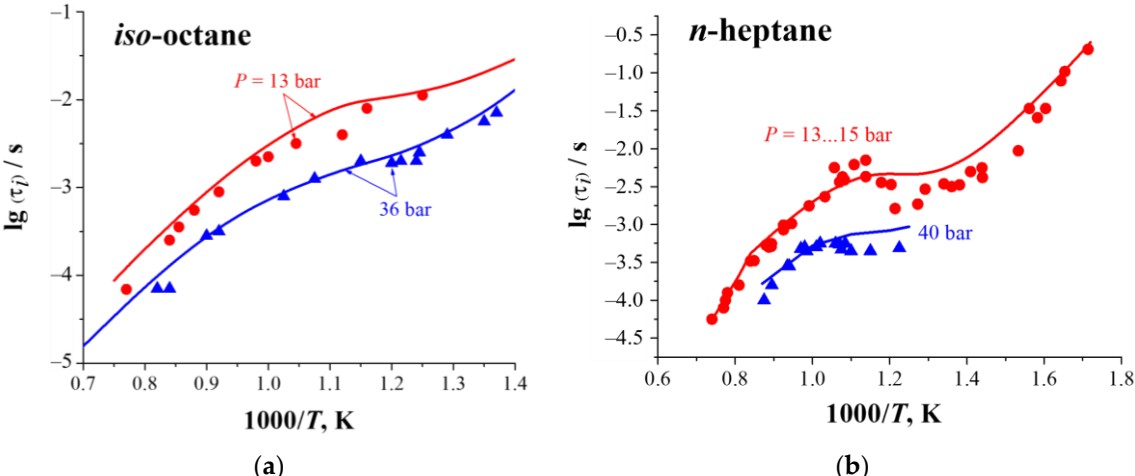

**Figure 2.** Comparison of predicted (curves) and measured (symbols) ignition delay times for the homogeneous stoichiometric iso-octane–air (**a**) and n-heptane–air (**b**) mixtures in wide ranges of initial conditions in terms of pressure and temperature: (**a**) circles, triangles [48]; (**b**) circles, triangles [49].

*2.2. Droplet Autoignition and Combustion*

The model of droplet autoignition was reported in [33,34]. The model is built on the concept of multicomponent diffusion in the gas, the DKM of hydrocarbon fuel oxidation, and the constant pressure condition in the gas-droplet system. The eminent feature of the used DKM is that it describes both multistage low-temperature oxidation with cool and blue flames and high-temperature droplet combustion. In the DKM, soot is designated as the C atom and is modeled by an equivalent gas species with a molecular mass of carbon, 12 kg/kmol. Herein, this model is only briefly presented. The set of governing equations includes:

The continuity equation for the liquid ($0 < r < r_m$):

$$\frac{\partial \rho_d}{\partial t} + \frac{1}{r^2}\frac{\partial}{\partial r}\left(r^2 \rho_d u_d\right) = 0 \tag{4}$$

where $r_m$ is the coordinate of the droplet surface (droplet radius); $t$ is time; $\rho$ is the density; $u$ is the velocity; and index $d$ denotes the liquid parameters;

The energy conservation equation for the liquid ($0 < r < r_m$):

$$c_d \rho_d \frac{\partial T_d}{\partial t} + c_d \rho_d u_d \frac{\partial T_d}{\partial r} = \frac{1}{r^2}\frac{\partial}{\partial r}\left(\lambda_d r^2 \frac{\partial T_d}{\partial r}\right) \tag{5}$$

$$T_d(0, r) = T_{d0}, \quad \left.\frac{\partial T_d}{\partial r}\right|_{r=0} = 0,$$

$$T_d(t, r_m) = T_g(t, r_m)$$

where $T_d(t, r)$ is the liquid temperature; $c_d(T_d)$ is the specific heat capacity of the liquid; $\lambda_d$ is the thermal conductivity of the liquid; index 0 denotes initial values; and index $g$ denotes gas properties;

The equation for the mass fraction of liquid vapor at the droplet surface ($r = r_m$):

$$Y_v = \frac{P_v}{P}\frac{W_v}{\overline{W}} \tag{6}$$

where $P$ is the pressure; $W$ is the molecular mass; index $v$ refers to the liquid vapor; and the overbar denotes the average value;

The continuity equation for the gas ($r_m < r < R_a$):

$$\frac{\partial \rho_g}{\partial t} + \frac{1}{r^2}\frac{\partial}{\partial r}\left(r^2\rho_g u_g\right) = 0 \tag{7}$$

$$\rho_d\left(u_d - \frac{\partial r_m}{\partial t}\right)\bigg|_{r=r_m} = \rho_g\left(u_g - \frac{\partial r_m}{\partial t}\right)\bigg|_{r=r_m}$$

where $R_a$ is the radius of the computational domain; and $\partial r_m/\partial t = u_m$ is the instantaneous velocity of the droplet surface due to both thermal expansion and evaporation;

The equation of continuity for gas species ($r_m < r < R_a$):

$$\rho_g\frac{\partial Y_j}{\partial t} + \rho_g u_g\frac{\partial Y_j}{\partial r} = \frac{1}{r^2}\frac{\partial}{\partial r}\left(\rho_g r^2 Y_j V_j\right) + \omega_{gj}$$

$$Y_j(0,r) = Y_{j0}, j = 1, 2, \ldots, N, \tag{8}$$

$$-\rho_d u_d \beta_i|_{r=r_m} = \rho_g Y_j\left(u_g - \frac{\partial r_m}{\partial t}\right) + \rho_g Y_j V_j\bigg|_{r=r_m},$$

$$\frac{\partial \overline{W} Y_j}{\partial r}\bigg|_{r=R} = 0, j = 1, 2, \ldots, N$$

where $V_j$ is the diffusion velocity of the $j$th species. The rates of chemical reactions $\omega_{gj}$ and the coefficients $\beta_i$ are determined as

$$\omega_{gj} = W_{gj}\sum_{k=1}^{L}\left(v''_{j,k} - v'_{j,k}\right)A_k T_g^{n_k}\exp\left(-\frac{E_k}{RT_g}\right)\prod_{l=1}^{N}\left(\frac{Y_{gl}\rho_g}{W_{gl}}\right)^{v'_{l,k}} \tag{9}$$

$$\beta_i = 1 \text{ at } j = v$$

$$\beta_i = 0 \text{ at } j \neq v$$

where $v'_{j,k}$ and $v''_{j,k}$ are the stoichiometric coefficients for the $j$th species in the case when it is a reactant and product in the $k$th reaction, respectively; $A_k$, $n_k$, and $E_k$ are the preexponential factor, temperature exponent, and activation energy for the $k$th reaction;

The equation for the diffusion velocity for the gas ($r_m < r < R_a$):

$$\frac{\partial X_j}{\partial r} = \sum_{k=1}^{N}\left(\frac{X_j X_k}{D_{jk}}\right)(V_k - V_j) \tag{10}$$

where $X_j = Y_j\overline{W}/W_j$ is the mole fraction of the $j$th component in the mixture;

The equations of conservation of energy for the gas ($r_m < r < R_a$):

$$c_{pg}\rho_g\frac{\partial T_g}{\partial t} + c_{pg}\rho_g u_g\frac{\partial T_g}{\partial r} = \frac{1}{r^2}\frac{\partial}{\partial r}\left(\lambda_g r^2\frac{\partial T_g}{\partial r}\right) + \Omega - \sigma S_{\text{rad}}Y_s\rho_g T_g^4 \tag{11}$$

$$T_g(0,r) = T_{g0}(r), T_g(t,r_m) = T_d(t,r_m), \frac{\partial T_g}{\partial r}\bigg|_{r=R} = 0$$

where $\rho_g = \rho_g(p, T_g)$, $c_{pg} = c_{pg}(T_g)$, and $\lambda_g = \lambda_g(p, T_g)$ are, respectively, the density, specific heat capacity, and thermal conductivity of the gas mixture. The term $\Omega$ in Equation (11) is given by

$$\Omega = \sum_{k=1}^{L} H_k A_k T_g^{n_k}\exp\left(-\frac{E_k}{RT_g}\right)\prod_{j=1}^{N}\left(\frac{Y_{gj}\rho_g}{W_{gj}}\right)^{v'_{l,k}}$$

where $H_k$ is the thermal effect of the *k*th chemical reaction. The last term in Equation (20) represents the heat loss due to soot radiation, the subject of our primary interest herein. In this term, $\sigma$ is the Stefan–Boltzmann constant, $Y_s$ is the mass fraction of soot, $S_{\text{rad}} = 6/(d_s\rho_s)$ is the specific surface area of conditional soot particles (here, $d_s$ is the conditional soot particle size, $\rho_s$ is the soot density). If one assumes $d_s \sim 1$ nm and $\rho_s \approx 2000$ kg/m$^3$, then $S_{\text{rad}} \approx 3 \times 10^6$ m$^2$/kg [50].

The condition at the droplet surface $(r = r_m)$ for determining the droplet surface temperature $T_{d,m}$:

$$\lambda_d \frac{\partial T_d}{\partial r} - \frac{\rho_{d,m} u_{d,m} L_v}{W_v} = \lambda_g \frac{\partial T_g}{\partial r} \tag{12}$$

where $L_v$ is the latent heat of liquid vaporization. Equation (12) is used for matching Equations (5) and (11);

The ideal gas EoS for the gas:

$$\rho_g = \frac{P\overline{W}}{RT_g} \tag{13}$$

The condition of constant pressure:

$$P = const \tag{14}$$

The set of Equations (4)–(14) allows one to determine the spatial structure of the flow around the droplet and its evolution in time, and to calculate the time dependences of the droplet diameter $d = 2r_m$, droplet surface temperature $T_{d,m}$, the maximum gas temperature $T_m = T_{g,max}$ (flame temperature), etc.

Figures 3–5 demonstrate the predicting capabilities of the model. Figure 3 compares the predicted and measured [51,52] time histories of the normalized squared diameter of n-heptane droplets at vaporization (Figure 3a) and combustion (Figure 3b). Figure 4 compares the predicted and measured [53–59] dependences of the combustion rate constant $K$ on the initial diameter of the n-heptane droplet in normal pressure and temperature conditions. Figure 5 compares the predicted and measured [60,61] ignition delays of single n-heptane droplets at temperatures of 940–1000 K and at different pressures. In general, the agreement between calculations and measurements is encouraging, keeping in mind that the accuracy of optical measurements of the droplet diameter is moderate, while the accuracy of ignition delay measurements can attain 100% [61].

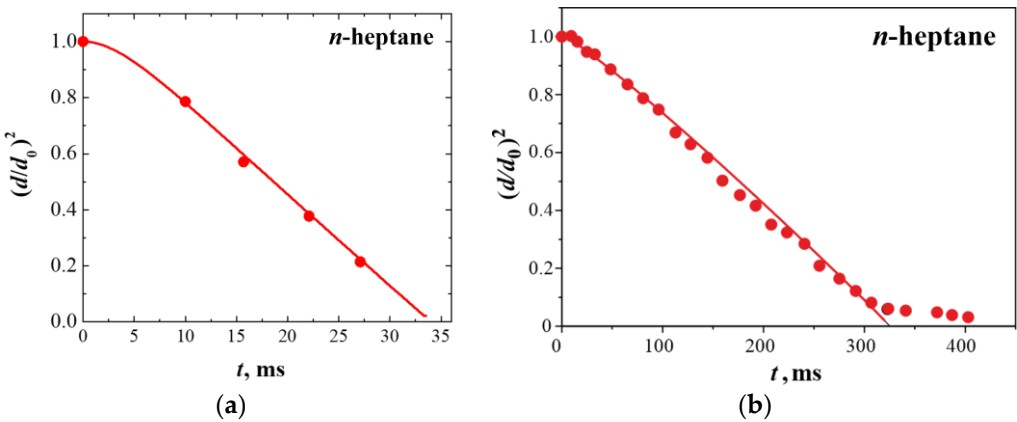

**Figure 3.** Comparison of predicted (curves) and measured (symbols) dynamics of n-heptane droplet vaporization and combustion in air: (**a**) vaporization, $d_0 = 70$ mm, $T_{d0} = 293$ K, $T_{g0} = 573$ K [51]; (**b**) combustion, $d_0 = 500$ mm, $T_{d0} = 293$ K, $T_{g0} = 293$ K [52].

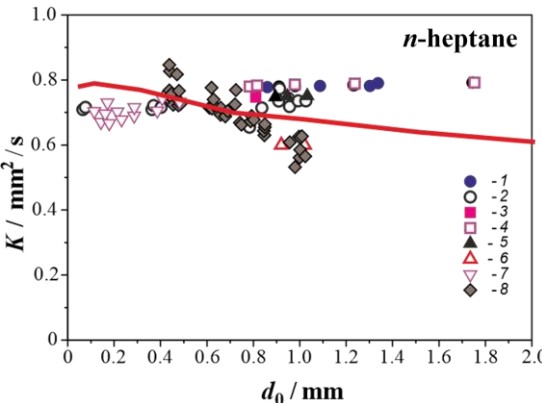

**Figure 4.** Comparison of predicted (curve) and measured (symbols) dependences of the combustion rate constant $K$ on the initial diameter of the n-heptane droplet. Experimental data: 1—[53], 2—[54], 3—[55], 4, 5—[56], 6—[57], 7—[58], 8—[59].

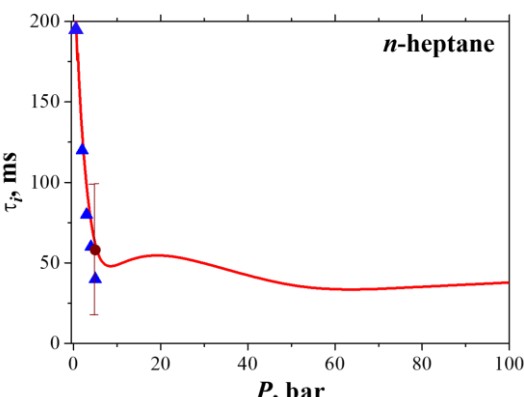

**Figure 5.** Comparison of predicted (curve) and measured (symbols) dependences of the autoignition delay time of n-heptane droplets on pressure. The curve is obtained for $d_0 = 700$ μm, $T_{d0} = 293$ K, and $T_{g0} = 1000$ K. Triangles correspond to $d_0 = 700$–$750$ μm [60], circle to $d_0 = 700$ μm, $T_{g0} = 940$ K [61] (the vertical bar shows the scatter of experimental data).

### 2.3. Real Gas Equation of State

The real gas thermal EoS is commonly written in a "virial" form:

$$\frac{Pv}{RT} = 1 + \frac{B}{v} + \frac{C}{v^2} + \frac{D}{v^3} + \frac{E}{v^4} + \frac{F}{v^5} + \ldots \tag{15}$$

where $B$, $C$, $D$, $E$, and $F$, ... are the virial coefficients depending on temperature $T$; $P$ is the pressure; $R$ is the universal gas constant, and $v$ is the molar volume. To describe the operation process in a diesel engine, we apply a relatively accurate real gas EoS proposed in [62]:

$$P = \rho RT \left[ 1 + B(T)\rho + b^2\rho^2 + b^3\rho^3 \right] \tag{16}$$

where $B(T)$ and $b$ are the coefficients of the truncated virial series. To determine the thermodynamic functions of a real gas, the concept of excess thermodynamic functions is used. According to this concept, the excess enthalpy $H_{exc}$ and excess internal energy $E_{exc}$ are given by:

$$H_{exc}(T,\rho) = E_{exc}(T,\rho) + P_{exc}(T,\rho)/\rho \tag{17}$$

$$E_{exc}(T,\rho) = -\int_0^\rho \left[ T(\partial p/\partial T)_\rho - P \right] d\rho/\rho^2 \tag{18}$$

where $\rho = 1/v$ is the molar density; and $P_{exc}$ is the excess pressure:

$$P_{exc} = P - P_0 \tag{19}$$

with index 0 denoting the ideal gas properties.

The enthalpy of a real gas is equal to:

$$H(T,\rho) = H_0(T) + H_{exc}(T,\rho) \tag{20}$$

where $H_0(T)$ is the ideal gas enthalpy; and $H_{exc}(T,\rho)$ is the excess enthalpy. The differentiation of Equation (7) with respect to temperature makes it possible to obtain an expression for the specific heat capacity of a real gas at a constant pressure, $C_p$:

$$C_p(T,\rho) = C_{p0}(T) + C_{p,exc}(T,\rho) \tag{21}$$

where $C_{p,exc}$ is the excess heat capacity at constant pressure, and $C_{p0}(T)$ is the specific heat capacity of the ideal gas.

The ideal gas thermal and caloric EoS are written in the standard form:

$$P_0 v = RT \tag{22}$$

$$dE_0 = C_{v0}(T)dT$$

$$dH_0 = C_{p0}(T)dT$$

where $E_0(T)$ is the specific molar internal energy and $C_{v0}$ is the specific heat capacity at constant volume. In accordance with Equation (19), in a real gas, there is a relative excess pressure equal to

$$P_{exc}/P_0 = (P - P_0)/P_0 = 1 + B(T)\rho + b^2\rho^2 + b^3\rho^3 \tag{23}$$

In [62], the values of coefficients $B(T)$ and $b$ are obtained for n-alkanes from methane to tetradecane, as well as for gases such as $O_2$, $N_2$, $H_2O$, CO, $CO_2$ and $H_2$ in wide ranges of pressure (from 0.5 to 200 bar) and temperatures (from 280 to 3000 K). As an example, Table 1 demonstrates the accuracy of Equation (16) for n-hexane and other mentioned gases at some selected isobars and isotherms. It turns out that the error in calculating the pressure according to Equation (16) does not normally exceed tenths of a percent, even in the vicinity of the critical point. Note that Equation (16) is used in the gas-dynamic calculation to determine the density $\rho$ from the known values of pressure $P$ and temperature $T$, as well as from the known composition of the gas mixture. To solve Equation (16) with respect to density, one has to use numerical methods.

**Table 1.** Comparison of predicted pressure, $P_{calc}$, given by the EoS of Equation (16) with measured pressure, $P_{exp}$, for n-hexane, oxygen, nitrogen, water, carbon monoxide, carbon dioxide, and hydrogen at some selected isobars and isotherms.

| | | **n-Hexane** | | |
|---|---|---|---|---|
| $T$, K | $\rho$, mol/dm$^3$ | $P_{calc}$, MPa | $P_{exp}$, MPa [63] | $\left\|P_{calc} - P_{exp}\right\|/P_{exp}$, % |
| 530 | 2.526 | 4.010 | 4 | 0.25 |
| 550 | 1.619 | 4.032 | 4 | 0.79 |
| 600 | 1.106 | 4.008 | 4 | 0.19 |
| 630 | 0.9741 | 4.0025 | 4 | 0.06 |

**Table 1.** *Cont.*

| Oxygen | | | | |
|---|---|---|---|---|
| $T$, K | $\rho$, kg/m$^3$ | $P_{\text{calc}}$, MPa | $P_{\text{exp}}$, MPa [64] | $|P_{\text{calc}} - P_{\text{exp}}|/P_{\text{exp}}$, % |
| 500 | 45.6 | 6.0008 | 6 | 0.013 |
| 500 | 60.51 | 8.0033 | 8 | 0.041 |
| 500 | 75.25 | 10.007 | 10 | 0.07 |
| 500 | 111.31 | 15.028 | 15 | 0.19 |
| 500 | 146.15 | 20.076 | 20 | 0.38 |
| Nitrogen | | | | |
| $T$, K | $\rho$, mol/dm$^3$ | $P_{\text{calc}}$, MPa | $P_{\text{exp}}$, MPa [65] | $|P_{\text{calc}} - P_{\text{exp}}|/P_{\text{exp}}$, % |
| 500 | 0.94635 | 4.0007 | 4 | 0.018 |
| 500 | 1.4070 | 6.0011 | 6 | 0.018 |
| 500 | 1.8590 | 8.0018 | 8 | 0.023 |
| 500 | 2.3020 | 10.002 | 10 | 0.022 |
| 500 | 3.3700 | 15.004 | 15 | 0.027 |
| 500 | 4.3806 | 20.004 | 20 | 0.022 |
| Water | | | | |
| $T$, °C | $v$, dm$^3$/g | $P_{\text{calc}}$, MPa | $P_{\text{exp}}$, MPa [66] | $|P_{\text{calc}} - P_{\text{exp}}|/P_{\text{exp}}$, % |
| 300 | 5.885 | 4.0066 | 4 | 0.17 |
| 300 | 4.532 | 5.017 | 5 | 0.34 |
| 300 | 3.616 | 6.033 | 6 | 0.55 |
| 300 | 2.976 | 7.0052 | 7 | 0.07 |
| 300 | 2.425 | 8.090 | 8 | 1.1 |
| Carbon monoxide | | | | |
| $T$, K | $\rho$, mol/dm$^3$ | $P_{\text{calc}}$, MPa | $P_{\text{exp}}$, MPa [67] | $|P_{\text{calc}} - P_{\text{exp}}|/P_{\text{exp}}$, % |
| 500 | 0.94818 | 4.004 | 4 | 0.09 |
| 500 | 1.40962 | 6.002 | 6 | 0.04 |
| 500 | 1.86196 | 7.999 | 8 | 0.01 |
| 500 | 2.30518 | 9.994 | 10 | 0.06 |
| 500 | 2.73932 | 11.990 | 12 | 0.08 |
| 500 | 3.16448 | 13.988 | 14 | 0.09 |
| 500 | 3.58073 | 15.989 | 16 | 0.07 |
| 500 | 3.98813 | 17.994 | 18 | 0.03 |
| 500 | 4.38673 | 20.004 | 20 | 0.02 |
| Carbon dioxide | | | | |
| $T$, °C | $\rho$, g/cm$^3$ | $P_{\text{calc}}$, MPa | $P_{\text{exp}}$, MPa [68] | $|P_{\text{calc}} - P_{\text{exp}}|/P_{\text{exp}}$, % |
| 300 | 56.42 | 6.000 | 6 | 0.00 |
| 300 | 75.59 | 8.000 | 8 | 0.01 |
| 300 | 94.89 | 10.000 | 10 | 0.00 |
| 300 | 114.26 | 12.000 | 12 | 0.00 |
| 300 | 133.67 | 14.001 | 14 | 0.01 |
| 300 | 153.09 | 16.006 | 16 | 0.04 |
| 300 | 172.4 | 18.009 | 18 | 0.05 |
| 300 | 191.6 | 20.014 | 20 | 0.07 |
| Hydrogen | | | | |
| $T$, K | $\rho$, mol/dm$^3$ | $P_{\text{calc}}$, MPa | $P_{\text{exp}}$, MPa [69] | $|P_{\text{calc}} - P_{\text{exp}}|/P_{\text{exp}}$, % |
| 500 | 0.94818 | 3.998 | 4 | 0.05 |
| 500 | 1.40962 | 5.994 | 6 | 0.10 |
| 500 | 1.86196 | 7.990 | 8 | 0.13 |
| 500 | 2.30518 | 9.986 | 10 | 0.14 |
| 500 | 2.73932 | 14.990 | 15 | 0.07 |
| 500 | 3.16448 | 20.03 | 20 | 0.15 |

For mixtures of real gases, the coefficients $B(T)$ and $b$ are calculated according to the following approximate mixing rules [62]:

$$B_{mix} = \sum_i^N B_i x_i; \quad b_{mix} = \sum_i^N b_i x_i \tag{24}$$

where $x_i$ is the volume fraction of substance $i$; and $N$ is the number of substances in the mixture.

The specific heat capacity at constant pressure is calculated by Equation (21). To determine the specific heat capacity of an ideal gas $C_{p0}$, a polynomial of the third order is used:

$$C_{p0} = a_1 + a_2 T + a_3 T^2 + a_4 T^3 \tag{25}$$

where $a_1, a_2, a_3$, and $a_4$ are tabulated coefficients [62]. The excess specific heat capacity $C_{p,\text{exc}}$ is calculated from analytical relationships that include the logarithmic derivatives of the coefficients $B(T)$ and $b$ with respect to temperature [62].

### 2.4. Solution Procedure of the Zero-Dimensional Problem

To reveal the possibility of modeling the multistage autoignition phenomena in CIEs with the DKM of iso-octane oxidation, zero-dimensional (0D) calculations of the kinetic processes occurring when a gas volume is compressed by a moving piston are performed using the CHEMKIN computer code [70]. Due to the uncertainty in estimates of the time-varying rate of heat transfer from the fuel charge to the engine wall, the velocity of gases, and turbulence, the process of gas cooling by the engine walls is not taken into account. Due to the short duration of the chemical process, the neglect of heat transfer cannot introduce a noticeable qualitative change in the calculations.

### 2.5. Solution Procedure of the One-Dimensional Problem

The set of Equations (4)–(11) is numerically integrated using a non-conservative implicit finite difference scheme and a moving adaptive mesh. The thermophysical properties of liquids and gases are taken from [48]. The chemical sources $\omega_{gj}$ and $\Omega$ are calculated using the DKM of n-heptane oxidation (available in the DKM of iso-octane oxidation) combined with the overall mechanism (OM) of soot formation [71]. The combined mechanism contains 623 reversible and 4 irreversible (Table 2) reactions and 84 species (including soot C). The parameters $A_k$ and $E_k$ in Table 2 are determined using the DKM of soot formation [72]. Contrary to the DKM of [72], in which soot nuclei are formed in processes involving a stable polyaromatic molecule and a radical or two polyaromatic radicals, the role of the soot precursor in the OM of Table 2 is attributed to acetylene $C_2H_2$. As an example, Figure 6 compares the soot yields predicted by the OM with those predicted by the DKM of [72] for the oxidation of homogeneous fuel-rich n-heptane–air mixture with a fuel-to-air equivalence ratio of $\Phi = 2$. Soot yield refers to the ratio of the mass of carbon contained in soot to the initial mass of carbon contained in the hydrocarbon fuel. As seen, the OM of Table 2 provides satisfactory qualitative and quantitative agreement with the results predicted by the DKM [72].

**Table 2.** Overall mechanism of soot formation.

| Reaction | $A_k$, (L, mol, s) | $E_k/R$, K | $n_k$ |
|---|---|---|---|
| $C_2H_2 + C_2H_2 = C + C + C_2H_4$ | $2 \times 10^{16}$ | 40,000 | 0 |
| $C + CO_2 = CO + CO$ | $1 \times 10^{15}$ | 40,000 | 0 |
| $C + H_2O = H_2 + CO$ | $1 \times 10^{15}$ | 40,000 | 0 |
| $C + OH = HCO$ | $1 \times 10^{12}$ | 0 | 0 |

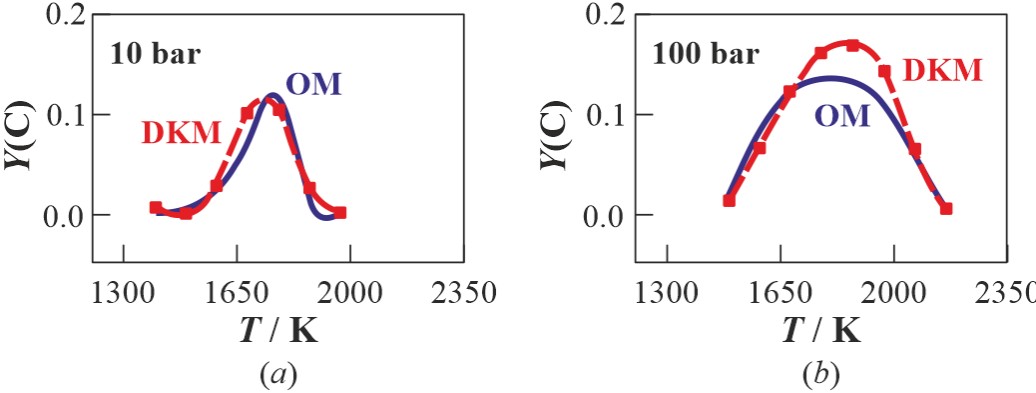

**Figure 6.** Soot yields $Y(C)$ during the oxidation of fuel-rich n-heptane–air mixture ($\Phi = 2$) as a function of temperature $T$ at $P = 10$ bar (**a**) and 100 bar (**b**) predicted by the DKM [72] and the OM of Table 1.

The calculation is based on the iterative procedure at each time step. The procedure involves the complete linearization of Equation (12) at the droplet surface. The solution accuracy is continuously monitored for compliance with the elemental balances of C and H atoms, as well as with the energy balance. The maximum deviation in the balances is 0.1%.

### 2.6. Solution Procedure of the Three-Dimensional Problem

To study the influence of real gas effects on the operation process of a diesel engine, the thermal and caloric EoS of [62] is implemented into the AVL FIRE code [73], which is widely used by engine companies worldwide for the design of reciprocating engines. By default, the code applies the ideal gas EoS. The object of the three-dimensional (3D) numerical study is a diesel engine with a semi-separated combustion chamber. Table 3 shows the parameters and conditions of the engine operation mode. This study is interested in the real gas effects on the indicator diagram and the yields of NO and soot.

**Table 3.** Parameters and conditions of the operation mode of the diesel engine *.

| Parameter | Value |
| --- | --- |
| Rotation speed, rpm | 2000 |
| Cylinder radius, mm | 42.5 |
| Compression ratio | 16 |
| Start of injection, CAD ** | 715.78 |
| End of injection, CAD | 730.06 |
| Injection angle, deg. | 150 |
| Mass of injected fuel, kg | $2.8 \times 10^{-5}$ |
| Fuel temperature, K | 330.15 |
| Mass fraction of exhaust gases | 0.233 |
| Equivalence ratio in exhaust gases | 0.5606 |
| Flow swirl, 1/min | 5800 |

* The data are provided by AVL LIST GmbH; ** CAD = Crank Angle Degree.

It is assumed that all cylinders of the engine operate in the same way, so only one engine cylinder is considered in the calculations. If we assume that the combustion process is axisymmetric, we can proceed to the consideration of a segment model, which is 1/8 of the combustion chamber in accordance with the number of nozzles in the injector, which greatly simplifies the geometry of the computational domain. However, such a simplification requires the account for the volume of technological recesses in the geometry of the combustion chamber associated with valves, etc. This is usually made by providing the so-called compensation volume (CV) in the geometry of the computational domain. The CV is an additional volume equal to the unaccounted volume of recesses in the piston and in the head cylinder block. In this case, the CV is placed along the cylinder wall. Figure 7

shows a base (Figure 7a) and fine (Figure 7b) computational meshes used in the calculations. The average cell size in the base and fine meshes is 0.5 and 0.25 mm, respectively. The computational mesh is movable.

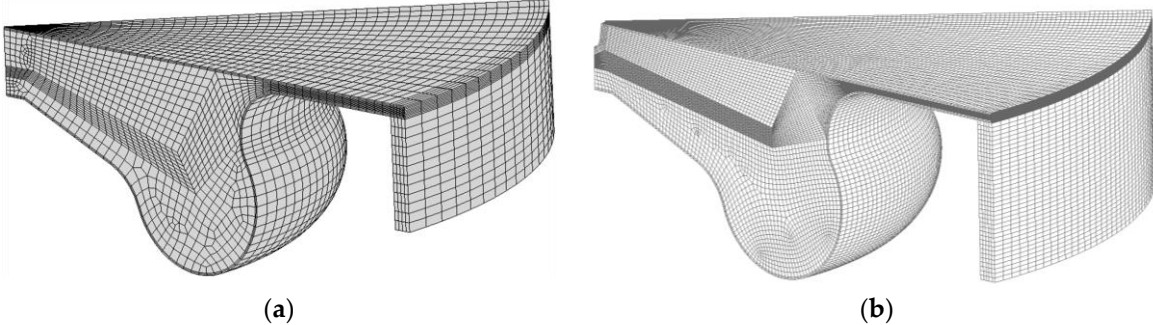

(**a**)          (**b**)

**Figure 7.** Base (**a**) and fine (**b**) computational meshes of the diesel segment at a top dead center (TDC); Base mesh contains 30,000 cells, average cell size 0.5 mm; Fine mesh contains 250,000 cells, average cell size 0.25 mm.

Numerical studies of the engine operation process are carried out on the basis of 3D Reynolds-averaged Navier–Stokes (RANS) equations supplemented by the $k$–$\xi$–$f$ turbulence model [74], the Lagrangian model of the fuel spray, and the three-zone combustion model ECFM- 3Z [75]. The motion, evaporation, and fragmentation of liquid droplets (diesel oil) in the fuel spray model are described by the standard Schiller–Neumann [76], Dukovich [77], and WAVE [78] models, respectively. For simplicity, it is assumed that the chemical surrogate of diesel oil is n-heptane possessing the cetane number ($\approx$56) close to that of diesel oil ($\approx$53) [79]. Thus, the gaseous combustible mixture may contain fuel vapor ($C_7H_{16}$), $O_2$, $N_2$, $CO_2$, $H_2O$, $CO$, and $H_2$. The formation of nitrogen oxides and soot is described by the Zel'dovich [80] and Kennedy–Hiroyasu–Magnussen [81] standard models. Due to the low concentrations, nitrogen oxides, soot, and active radicals present in the combustion model are not included in the material balance.

Periodic boundary conditions are set on the side surfaces of the segment. The walls of the combustion chamber are considered nonslip, isothermal (piston 475.15 K, head 450.15 K, cylinder walls 375.15 K), impermeable, and noncatalytic. To avoid excessive thickening of the computational mesh near the walls, the formalism of wall functions is used in the calculations. The calculation starts from a crank angle of 566.5 CAD and ends at 860 CAD.

To solve the system of governing mass, momentum, and energy conservation equations, a segregated algorithm of the SIMPLE type (semi-implicit method for pressure-linked equations) is used [82]. The convective transport in the mass conservation law is approximated by the central difference, in the momentum conservation law by the Total Variation Diminishing (TVD) scheme with the MINMOD limiter [83], and for the rest of the conservation equations, the standard first-order UPWIND scheme is used.

Figure 8 demonstrates the predictive capabilities of the model. It compares the measured pressure in the diesel engine under consideration with the pressure histories calculated using the base and fine meshes. In the calculations, the ideal-gas thermal and caloric EoS are used. To fit the measured pressure curve, the empirical constants of the WAVE model were slightly adjusted (within the recommended range) as compared to the default values ($C_1 = 0.61$ and $C_2 = 30$), but were the same for both meshes. Despite the use of the simplified engine model, the calculated pressure histories are seen to satisfactorily agree with each other and with the measurements. This comparison justifies the use of the base computational mesh for simulating the engine operation process.

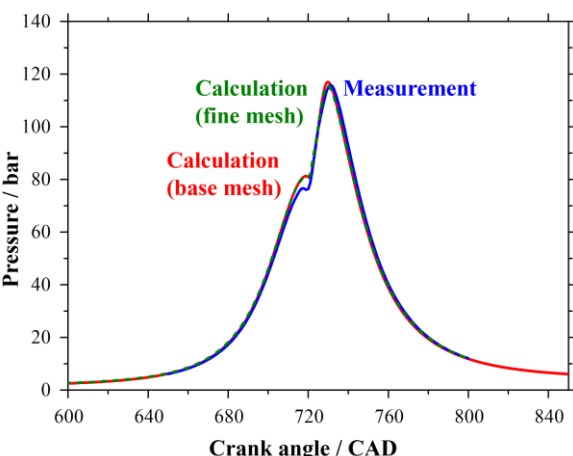

**Figure 8.** Calculated and measured time histories of the in-cylinder pressure.

## 3. Results and Discussion

### 3.1. Multistage Fuel-Oxidation Chemistry

Figure 9 presents the results of the 0D calculations in terms of the time histories of pressure $P$, temperature $T$, and volume fractions of OH, iso-octyl hydroperoxide ($CH_3C(CH_3)_2$ $CH_2CH(CH_3)CH_2O_2H$), and hydrogen peroxide ($H_2O_2$) in the engine cylinder at an air-to-fuel equivalence ratio $\alpha = 1.5$ ($\Phi = 0.667$) and different compression ratios $\varepsilon$: 12 (Figure 9a); 13 (Figure 9b); 13.5 (Figure 9c); 13.75 (Figure 9d); and 14 (Figure 9e). In all plots, the time is plotted along the X-axis in terms of the crankshaft rotation angle $\theta$ (in CAD). The extreme position of the piston (TDC, 0 CAD) is shown by the dashed vertical line. For the sake of convenience, the pressure $P$ (in bar), temperature $T$ (in K), and volume fractions of OH, $CH_3C(CH_3)_2CH_2CH(CH_3)CH_2O_2H$, and $H_2O_2$ are all plotted along the Y-axis with some scaling factors indicated near the corresponding curves. The mixture composition with $\alpha = 1.5$ ($\Phi = 0.667$) is chosen for calculations because the boundaries of single cool flame, "double" cool flame, blue flame, and hot flame domains for this mixture in the experiments are well separated (see the dashed vertical line in Figure 1). All other calculation conditions are the same as in the experiments.

At $\varepsilon \leq 12$ (see Figure 9a), the reaction is not noticeable in the calculations. The pressure and temperature curves are virtually symmetrical with respect to the dashed vertical line marking the TDC position. The experimental limit for a given mixture with $\alpha = 1.5$ corresponds to $\varepsilon \sim 9$.

At $\varepsilon = 13$ (see Figure 9b), the calculated pressure and temperature curves detect a deviation from the symmetrical shape starting from $\theta \sim 7$ CAD. This deviation corresponds to the heat release in a single cool flame. As a matter of fact, the volume fraction of OH has a peak at $\theta \sim 9$ CAD, where the rate of iso-octyl hydroperoxide decomposition attains the maximum value, while hydrogen peroxide is only accumulated. The experimental range of single cool flames at $\alpha = 1.5$ corresponds to $\varepsilon \sim 9$–15.

At higher compression ratios, the authors of [4] identified the occurrence of so-called "double" cool flames (see Figure 1). As a matter of fact, according to our calculations, the domain of "double" cool flames corresponds to the sequential appearance of a separated cool flame (arising due to the decomposition of iso-octyl hydroperoxide) and a low-intensity blue flame (arising due to the decomposition of $H_2O_2$), rather than two successive cool flames. In other words, in this parametric domain, the multistage nature of the low-temperature autoignition of hydrocarbons manifests itself. In laboratory conditions, multiple cool flames can only occur when strong cooling is applied to the reactor walls and the reaction progress is insignificant. In this case, almost identical conditions can be reproduced for restarting the cool-flame reactions. In the internal combustion engines, such conditions are not realized.

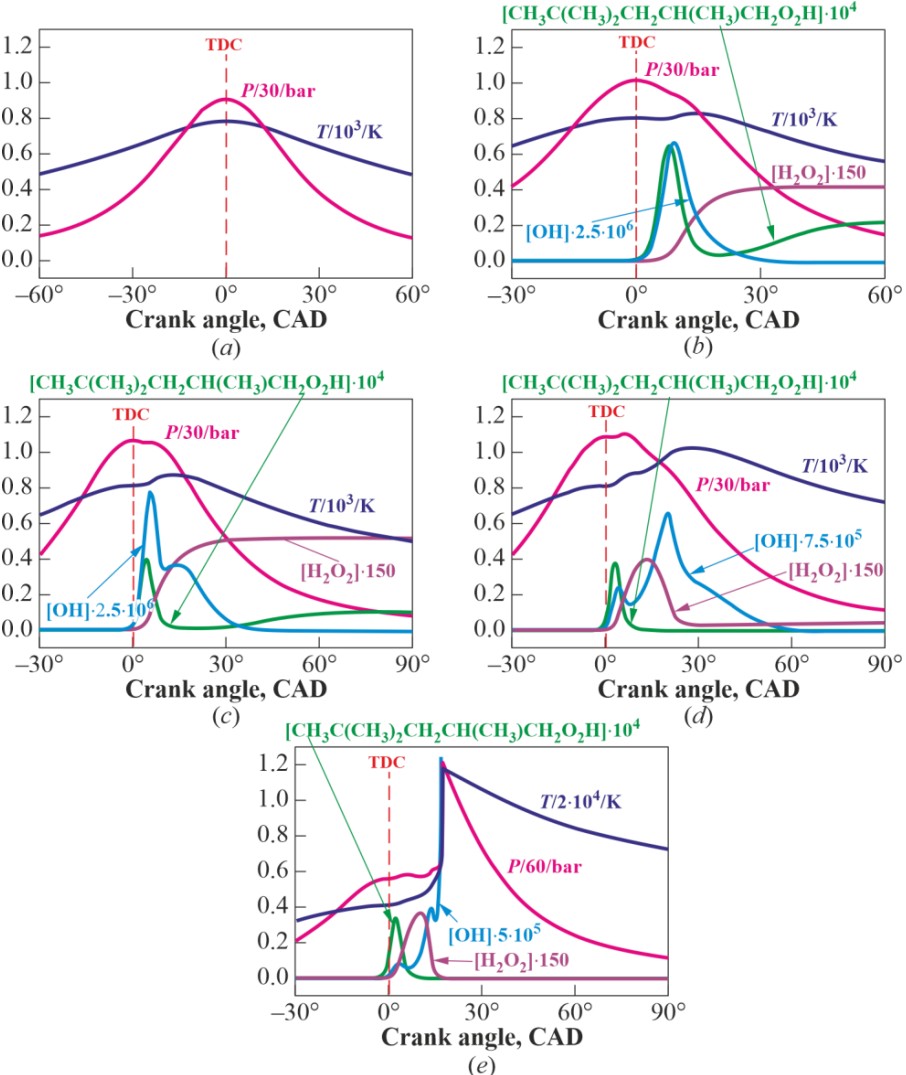

**Figure 9.** Calculated time histories of pressure $P$, temperature $T$, and volume fractions of OH, CH$_3$C(CH$_3$)$_2$CH$_2$CH(CH$_3$)CH$_2$O$_2$H, and H$_2$O$_2$ at an air-to-fuel equivalence ratio of 1.5 and different compression ratios $\varepsilon$: (**a**) 12; (**b**) 13; (**c**) 13.5; (**d**) 13.75; (**e**) 14.

In the calculations, the "double" cool flames are detected at $\varepsilon = 13.5$ in terms of a double peak of hydroxyl concentration during the expansion of the gas volume after TDC (see Figure 9c). On the pressure and temperature curves, the second peak is virtually not distinguished. The first (large) peak of OH corresponds to the maximum rate of iso-octyl hydroperoxide decomposition at $\theta \sim 6$ CAD. Thereafter, the second (small) peak of OH forms from $\theta \sim 7$–8 to $\sim 14$ CAD, when iso-octyl hydroperoxide still continues to decompose. Here, H$_2$O$_2$ is only accumulated, but, probably, the inverse process of H$_2$O$_2$ decomposition is already beginning to noticeably proceed.

The calculated case of clearly separated cool and blue flames corresponds to $\varepsilon = 13.75$ (see Figure 9d). There is also a double peak of OH, but at a significantly increased OH volume fraction (a scaling factor is reduced by a factor of 3). The first peak corresponds to the maximum rate of iso-octyl hydroperoxide decomposition at $\theta = 4.2$ CAD. The second (larger) peak corresponds to the maximum rate of decomposition of the accumulated hydrogen peroxide at $\theta = 20$ CAD. Here, the pressure and temperature are higher than in Figure 9a–c but have the same order of magnitude (the scale factor is retained), and there is no hot autoignition. With an increase in the compression ratio, an intense blue flame arises. The luminosity of the blue flame can clog the weaker luminosity of a cool flame, and then the overall luminosity looks like a true blue flame. Apparently, this is precisely why the

authors of [4] marked the lower blurred boundary of the blue flame domain with dashes (see Figure 1e). The experimental range of "double" cool flames for $\alpha = 1.5$ corresponds to $\varepsilon \sim 15$–17, while the range of blue flames corresponds to $\varepsilon \sim 17.5$–19.5.

Finally, at $\varepsilon = 14$ (see Figure 9e), hot autoignition is detected in the calculation. The pressure and temperature peaks are doubled (scale factors are reduced by a factor of 2), and the concentration peak of hydroxyl is orders of magnitude higher (the maximum is beyond the graph). In the experiments with $\alpha = 1.5$, the hot flame corresponds to the values $\varepsilon > 19.5$.

Table 4 additionally compares the results of calculations and experiments at different temperatures at the end of compression, $T_c$ (temperatures at TDC), corresponding to the chosen $\varepsilon$ values (for experimental values, we provide an estimate of the compression ratio $\varepsilon$). The comparison of results allows us to make a general conclusion that the calculations are in qualitative agreement with the experimental data.

**Table 4.** Calculated compression ratios $\varepsilon$ and temperatures $T_c$ at the end of compression ($T_0$ = 343 K, $P_0$ = 1 bar, $n$ = 1500 rpm, $\alpha$ = 1.5).

| Reaction Type | $\varepsilon$ | $T_c$, K | Exp. Range $T_c$, K |
|---|---|---|---|
| No apparent reaction | 12.00 | <786 | <703 |
| Single cool flames | 13.00 | 808 | 703–838 |
| Double cool flames | 13.50 | 816 | 838–882 |
| Blue flames | 13.75 | 820 | 882–914 |
| Hot flames | 14.00 | >826 | >914 |

*3.2. Droplet Autoignition*

The self-ignition of droplets is simulated by the instantaneous placement of a droplet in a uniformly heated gas with $T_{g0}$ = 1000 K. Below, attention is paid to the autoignition of small (submillimeter) n-heptane droplets with an initial temperature $T_{d0}$ = 293 K in air at pressures up to 100 bar.

Figure 10 shows the results of the calculations for droplets of initial diameter $d_0$ = 50, 100, and 200 mm as dependences of the maximum gas temperature $T_{max}$ on the reduced time $t/d_0^2$. For convenience, all curves are plotted in such a way that the autoignition of droplets occurs at time $t = 0$. Breaks in the curves at $t > 0$ correspond to the complete combustion of droplets. With a decrease in the initial droplet size from 200 to 50 μm, the effect of losses on thermal radiation decreases: during the lifetime of the droplet at 50 μm in initial diameter, the flame temperature decreases by only 300 K, from 2500 to 2200 K, whereas for the droplet 200 μm in initial diameter, the flame temperature decreases by about 550 K.

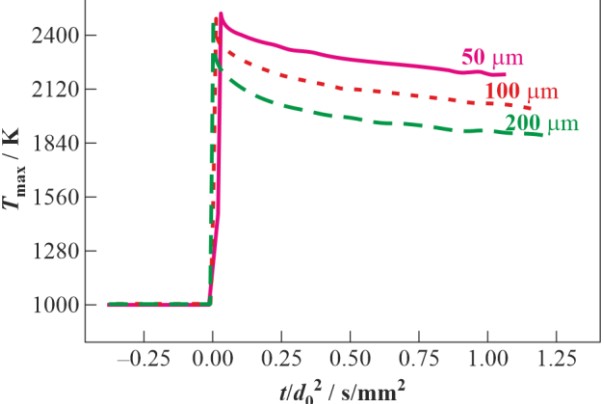

**Figure 10.** Calculated dependences of the maximum gas temperature in the vicinity of a droplet on the reduced time during the autoignition of droplets of different initial diameters (50 mm; 100 mm; and 200 mm) in air at $T_{g0}$ = 1000 K and $P$ = 100 bar.

Figure 11 shows the instantaneous spatial distributions of temperature and the mass fraction of soot around a droplet with $d_0 = 50$ μm at $t = 1.5$, 2, and 3 ms. The dimensionless radial distance $2r/d_0$ from the droplet center is plotted along the abscissa. The temperature distribution around the droplet has a dome shape with a pronounced maximum corresponding to the combustion temperature. The distribution of the mass fraction of soot also has a dome shape, and the maximum of this distribution is between the surface of the droplet and the flame, i.e., in the region where fuel vapor is accumulated, and molecular oxygen in the air is absent. The maximum temperature is seen to be attained at a distance up to $2d_0$ from the droplet surface, while the maximum mass fraction of soot is seen to be reached at a distance of $\sim 0.5d_0$. The characteristic width of the soot "shell" is $\sim d_0$.

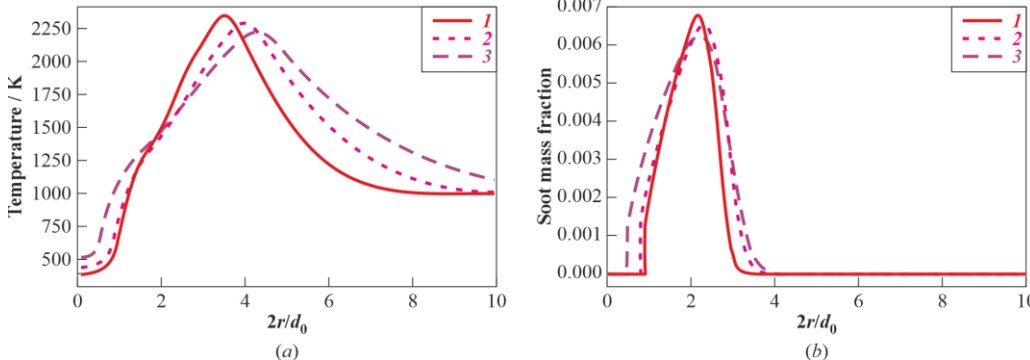

**Figure 11.** Calculated instantaneous spatial distributions of temperature (**a**) and soot mass fraction (**b**) around a droplet of initial diameter $d_0 = 50$ μm during its autoignition at $T_{g0} = 1000$ K and $P = 100$ bar: 1—$t = 1.5$ ms; 2—2 ms; 3—$t = 3$ ms.

To evaluate the effect of soot thermal radiation on the combustion of small (submillimeter size) droplets, a series of calculations were carried out with and without soot radiation taken into account. Figure 12 shows examples of the calculated dependences of the maximum gas temperature $T_{max}$ around single small droplets with a diameter of 20 (Figure 12a) and 40 μm (Figure 12b) during their two-stage autoignition and subsequent combustion with and without allowance for the thermal radiation of the formed soot. It can be seen that as $T_{max}$ increases to the maximum value, both curves in Figure 12a,b merge, and after reaching the maximum, they slightly diverge. The merging of the curves in the initial section is explained by the negligible intensity of heat losses due to radiation compared to the intensity of chemical energy release during the autoignition of the fuel–air mixture in the vicinity of the droplet. Heat losses due to radiation begin to manifest themselves only after $T_{max}$ reaches its maximum value and the rate of energy release decreases to a level characteristic of the diffusion combustion of the droplet.

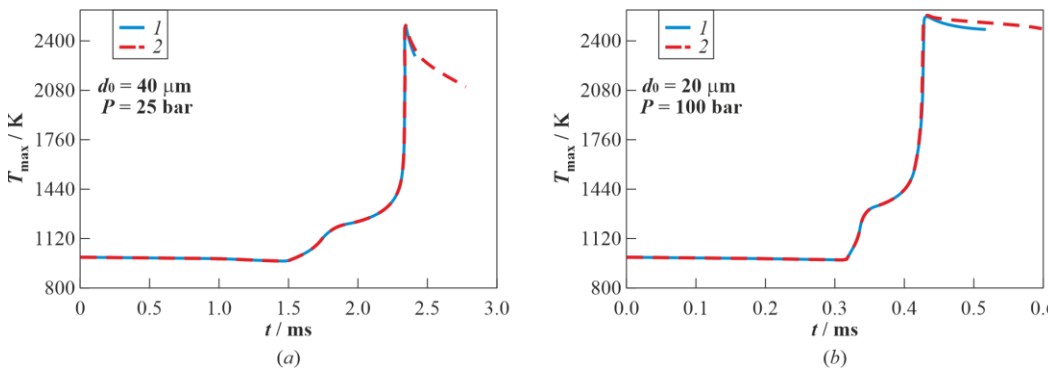

**Figure 12.** Calculated dependences of the maximum gas temperature $T_{max}$ around the n-heptane droplet during its autoignition and subsequent combustion; $T_{d0} = 293$ K; $T_{g0} = 1000$ K: (**a**) $d_0 = 40$ μm, $P = 25$ bar; (**b**) $d_0 = 20$ μm, $P = 100$ bar; 1—calculation with thermal radiation of soot; 2—calculation without thermal radiation of soot.

The weak effect of thermal radiation on the autoignition and subsequent combustion of small droplets is confirmed by the spatial temperature distributions around the droplets, plotted for time instants closely after autoignition (Figure 13), as well as by the time histories of the squared droplet diameter (Figure 14). As follows from Figures 13 and 14, taking into account losses due to radiation virtually has no effect on the temperature curves or on the value of the combustion rate constant *K*. Moreover, radiation losses only begin to manifest themselves when the droplets have almost completely evaporated.

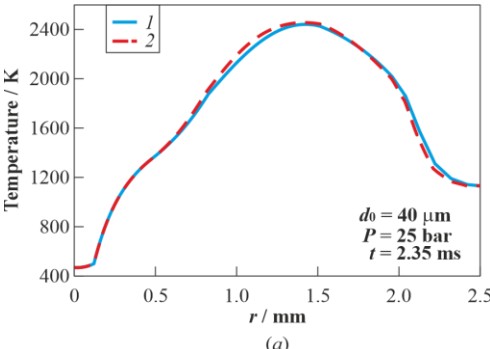
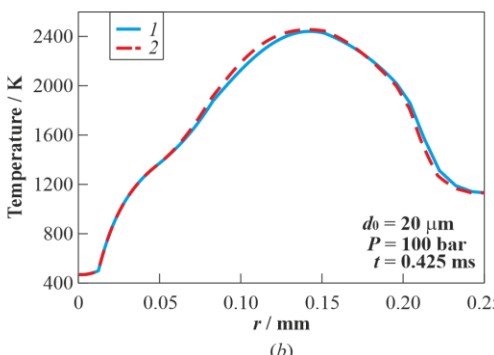

**Figure 13.** Estimated instantaneous temperature distributions in a droplet and in a gas depending on the distance from the droplet center; $T_{d0}$ = 293 K; $T_{g0}$ = 1000 K: (**a**) $d_0$ = 40 μm, $P$ = 25 bar, $t$ = 2.35 ms; (**b**) $d_0$ = 20 μm, $P$ = 100 bar, $t$ = 0.425 ms; 1—calculation with thermal radiation of soot; 2—calculation without thermal radiation of soot.

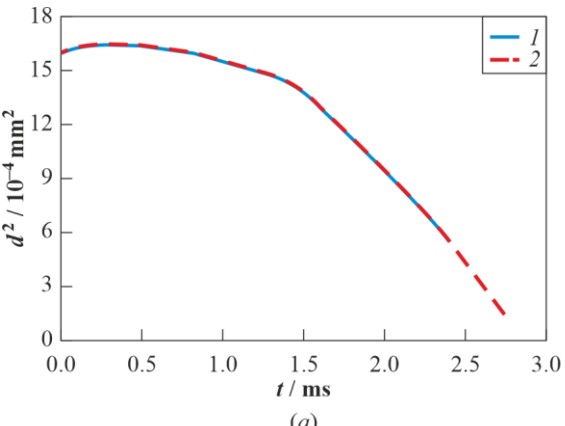
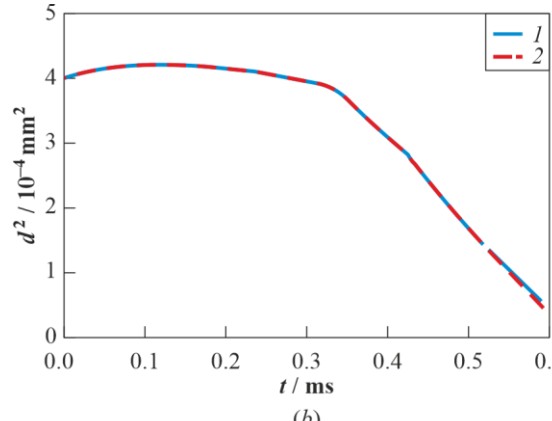

**Figure 14.** Calculated time histories of the squared droplet diameter during droplet autoignition and subsequent combustion; $T_{d0}$ = 293 K; $T_{g0}$ = 1000 K: (**a**) $d_0$ = 40 μm, $P$ = 25 bar; (**b**) $d_0$ = 20 μm, $P$ = 100 bar: 1—calculation with thermal radiation of soot; 2—calculation without thermal radiation of soot.

Thus, the effect of thermal radiation on soot during the autoignition and subsequent combustion of small droplets turns out to be weak: the droplets burn out before the effects of thermal radiation manifest themselves. The latter seems to be important when modeling the operation process in CIEs. The autoignition and combustion of droplets under such conditions can be approximately simulated without taking into account the thermal radiation of soot. In addition, it should be borne in mind that in the presence of directed convective flows in the vicinity of a droplet, the soot shell is deformed and fragmented, which also reduces the effect of soot thermal radiation on droplet combustion. Another implication of this study is that submillimeter-sized droplets of hydrocarbon fuels do not exhibit radiative extinction, contrary to large droplets that are several millimeters in size. The lifetime of such small droplets appears to be shorter than the characteristic time of radiative flame quenching.

### 3.3. Real Gas Effects

Three test calculations were carried out using the thermal and caloric EoS of real and ideal gases: (I) ideal gas thermal EoS of Equation (22) with the specific heat capacity of Equation (25); (II) real gas thermal EoS of Equation (16) with the specific ideal-gas heat capacity of Equation (25); and (III) real gas thermal EoS of Equation (16) with the specific heat capacity of Equation (21). All calculations were made using the base mesh of Figure 7a under completely identical model settings as well as initial and boundary conditions. Note, that the test calculation II is inherently incorrect and was only carried out to illustrate the important role played by the excess heat capacity in a real gas.

The calculations show the significant role of the real gas effects in the operation process of a diesel engine. Figure 15a,b compares three dependences of the pressure and mass-averaged temperature in the combustion chamber on the crank angle obtained in calculations I, II, and III. It follows from the comparison of curves I and III in Figure 15a,b that the account for the real gas properties decreases the maximum pressure and mass-averaged temperature in the combustion chamber by approximately 7 bar (6%) and 150 K (9%), respectively.

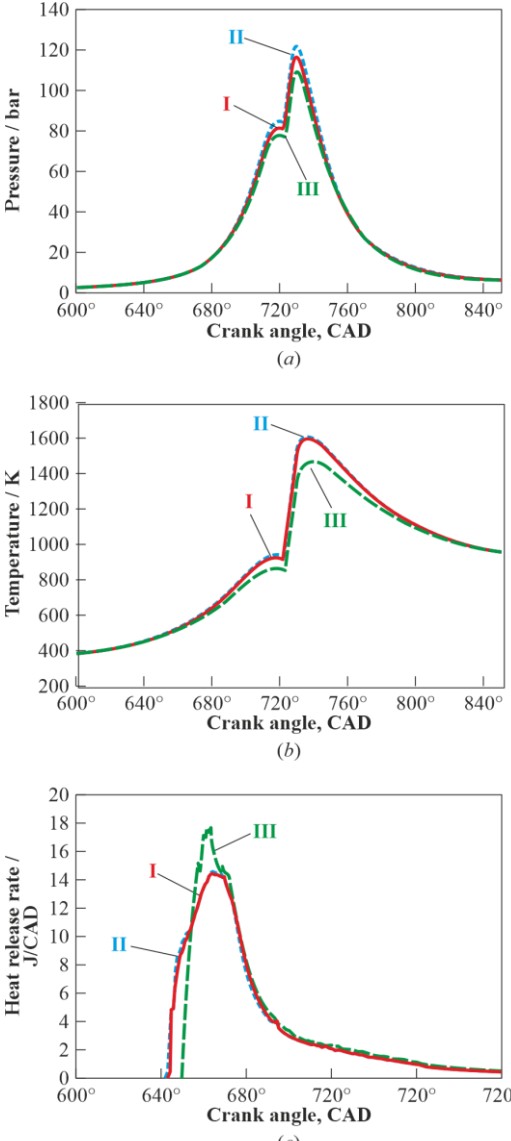

**Figure 15.** Predicted dependences of pressure (**a**), mass-averaged temperature (**b**), and the total heat release rate (**c**) in the diesel cylinder on the crank angle in test calculations I–III.

The autoignition delay time and the total rate of heat release in the combustion chamber also change. This is clearly seen in Figure 15c, which shows the dependences of the total heat release rate on the crank angle obtained in the three considered test calculations. Compared to calculation I, in calculation III, the autoignition delay increases by 1.6 CAD, and the maximum heat release rate increases by 20%: from 15 to 18 J/CAD.

An increase in the autoignition delay and the total heat release rate is associated with a change in the distributions of the most important flow parameters in the combustion chamber, in particular temperature. Figures 16 and 17 show the instantaneous temperature distributions in the combustion chamber obtained in calculations I and III at different instants of time: from 721 to 728 CAD. It can be seen that, in calculation III, the autoignition of fuel vapors occurs later (~724 CAD) than in calculation I (~722 CAD). In addition, the locations of autoignition centers also differ in these calculations.

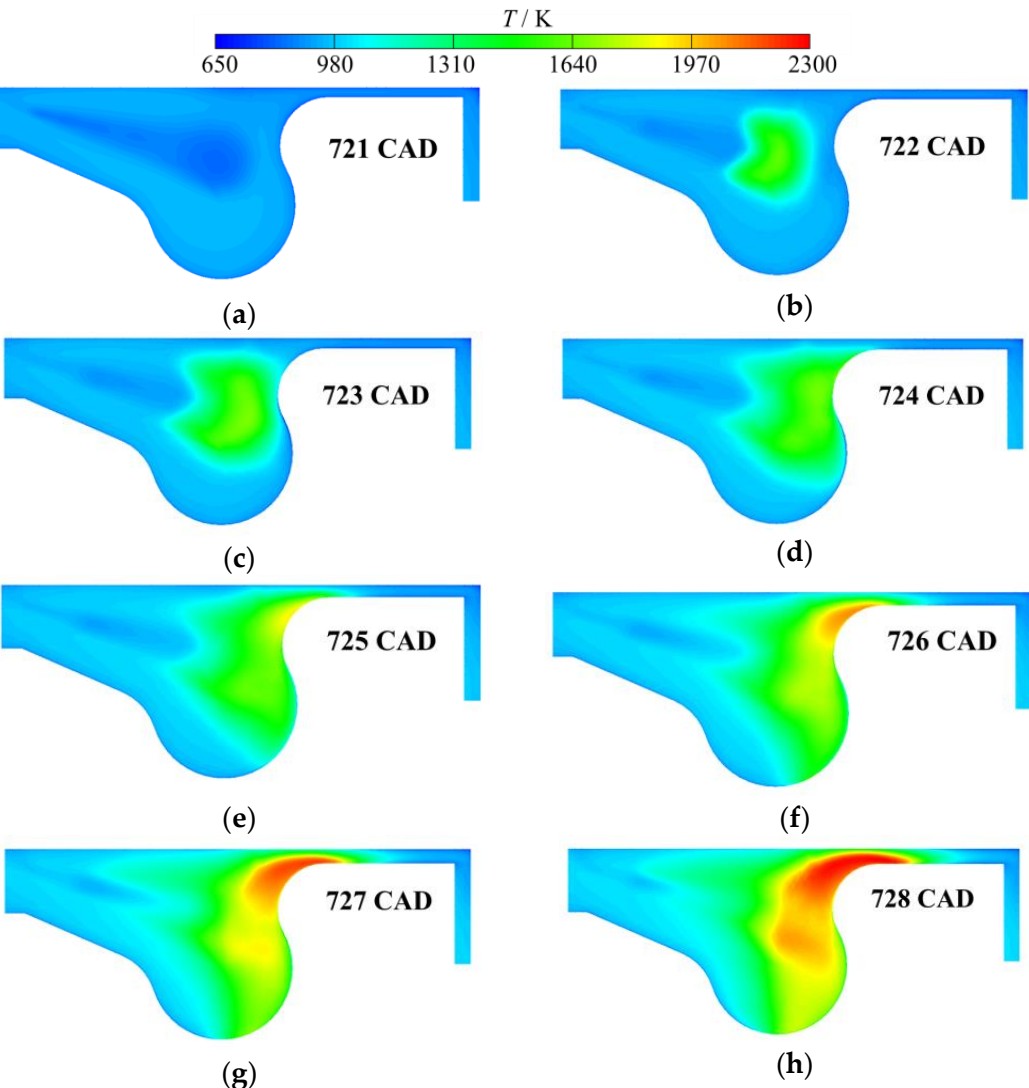

**Figure 16.** Predicted instantaneous temperature distributions in the combustion chamber at different instants of time obtained in calculation I: (**a**) 721 CAD; (**b**) 722 CAD; (**c**) 723 CAD; (**d**) 724 CAD; (**e**) 725 CAD; (**f**) 726 CAD; (**g**) 727 CAD; and (**h**) 728 CAD.

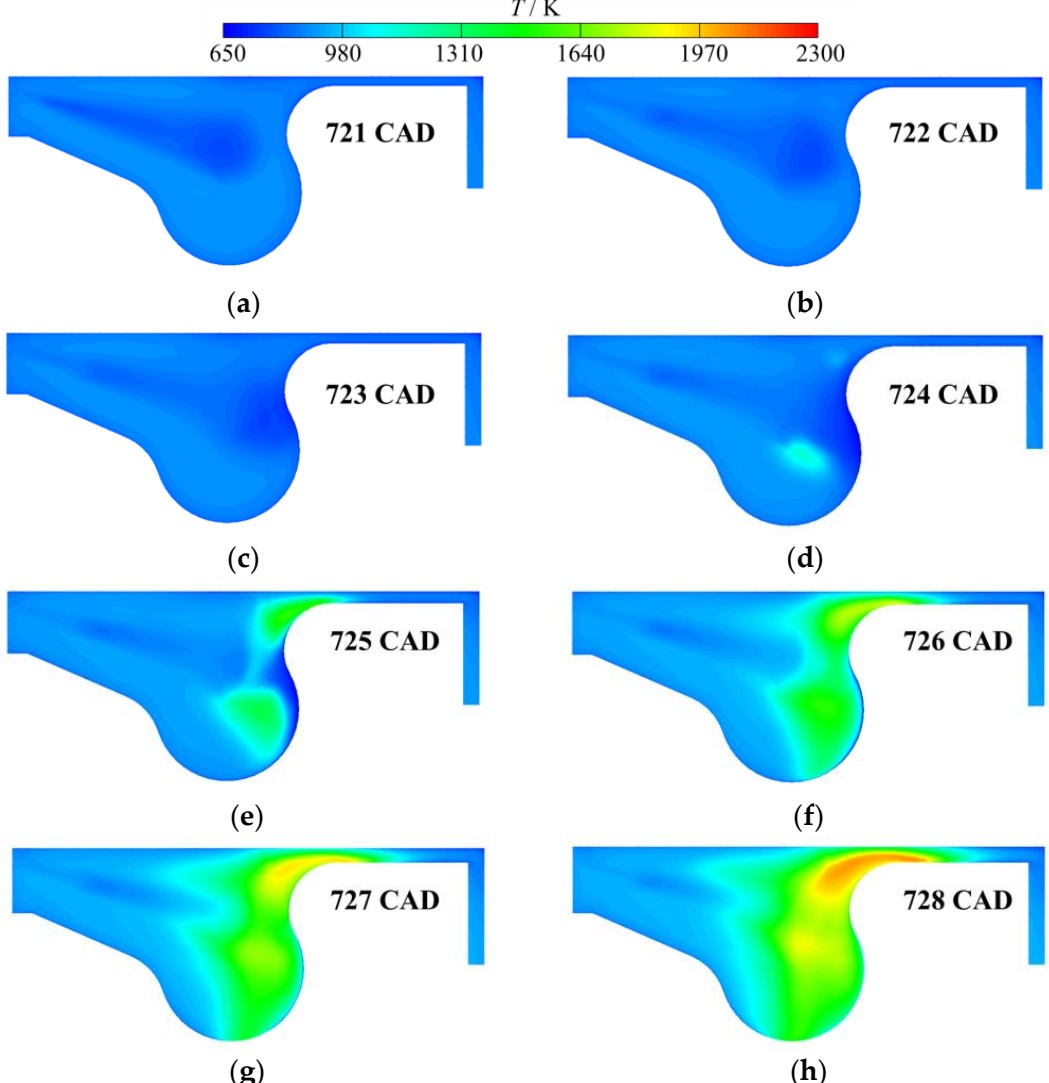

**Figure 17.** Predicted instantaneous temperature distributions in the diesel combustion chamber at different instants of time obtained in calculation III: (**a**) 721 CAD; (**b**) 722 CAD; (**c**) 723 CAD; (**d**) 724 CAD; (**e**) 725 CAD; (**f**) 726 CAD; (**g**) 727 CAD; and (**h**) 728 CAD.

The changes in the distributions of the flow parameters, autoignition delay time, and the maximum temperature levels in the combustion chamber caused by the account of real gas properties significantly affect the yields of NO and soot (Figure 18). Thus, in calculation III, the final values of NO and soot mass fractions turn out to be lower than in calculation I by a factor of 2 and 4, respectively. Moreover, in comparison with calculation I, an interesting detail is revealed in calculation III. In calculation I, soot forms in the vicinity of the cylinder wall (shown by an arrow in Figure 19), whereas in calculation III, soot does not form in this region. These differences are related to the reduced temperature of the real gas in the vicinity of the wall.

To understand in which zones of the combustion chamber real gas effects are mainly manifested, let us consider Figure 20. Figure 20 shows the spatial distributions of the relative excess heat capacity $C_{p,\text{exc}}/C_{p0}$ and the relative excess pressure $P_{\text{exc}}/P_0$ of Equation (6) obtained by calculation III. Prior to autoignition, the relative excess heat capacity of the medium in the combustor is, on average, 1–1.5%, while its maximum value (~2.5%) is achieved near the "cold" walls of the combustion chamber and in the region of the fuel spray with a high volume fraction of fuel vapors and a relatively low temperature; i.e., in those zones of the combustion chamber where the density of the matter is high. In hot

combustion products, the value of the relative excess heat capacity is about 0.5%. The average value of the relative excess pressure in the combustion chamber until the moment of autoignition is ~2.5%, whereas its maximum value is also reached near the "cold" walls of the combustion chamber and in the liquid fuel spray (~3%). After the autoignition of the mixture and subsequent increase in pressure, the maximum value of the relative overpressure reaches 4%, and in hot combustion products, it does not exceed 2%.

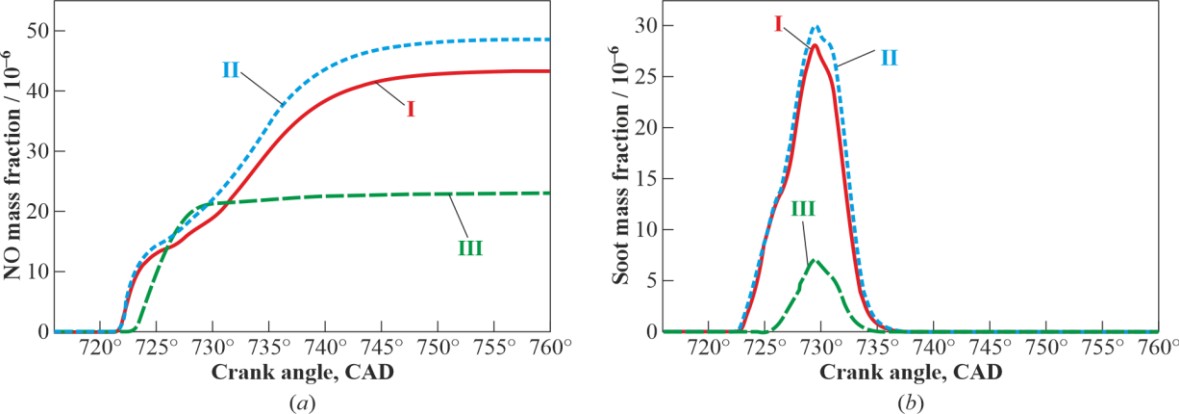

**Figure 18.** Predicted dependences of the mass fractions of nitrogen oxides (**a**) and soot (**b**) in the combustion chamber on the crank angle obtained in calculations I–III.

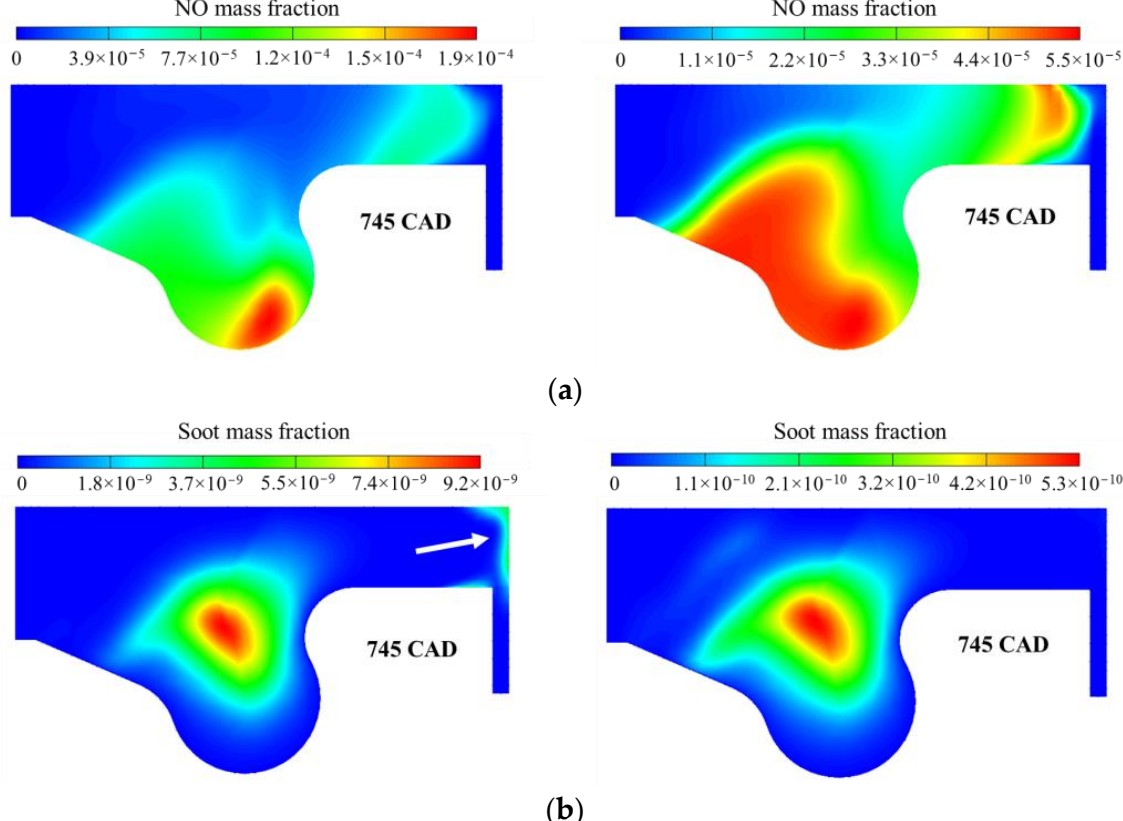

**Figure 19.** Predicted instantaneous distributions of the mass fractions of NO (**a**) and soot (**b**) in the diesel combustion chamber obtained in calculations I (left column) and III (right column) at 745 CAD. White arrow shows the location in the vicinity of the cylinder wall where soot forms.

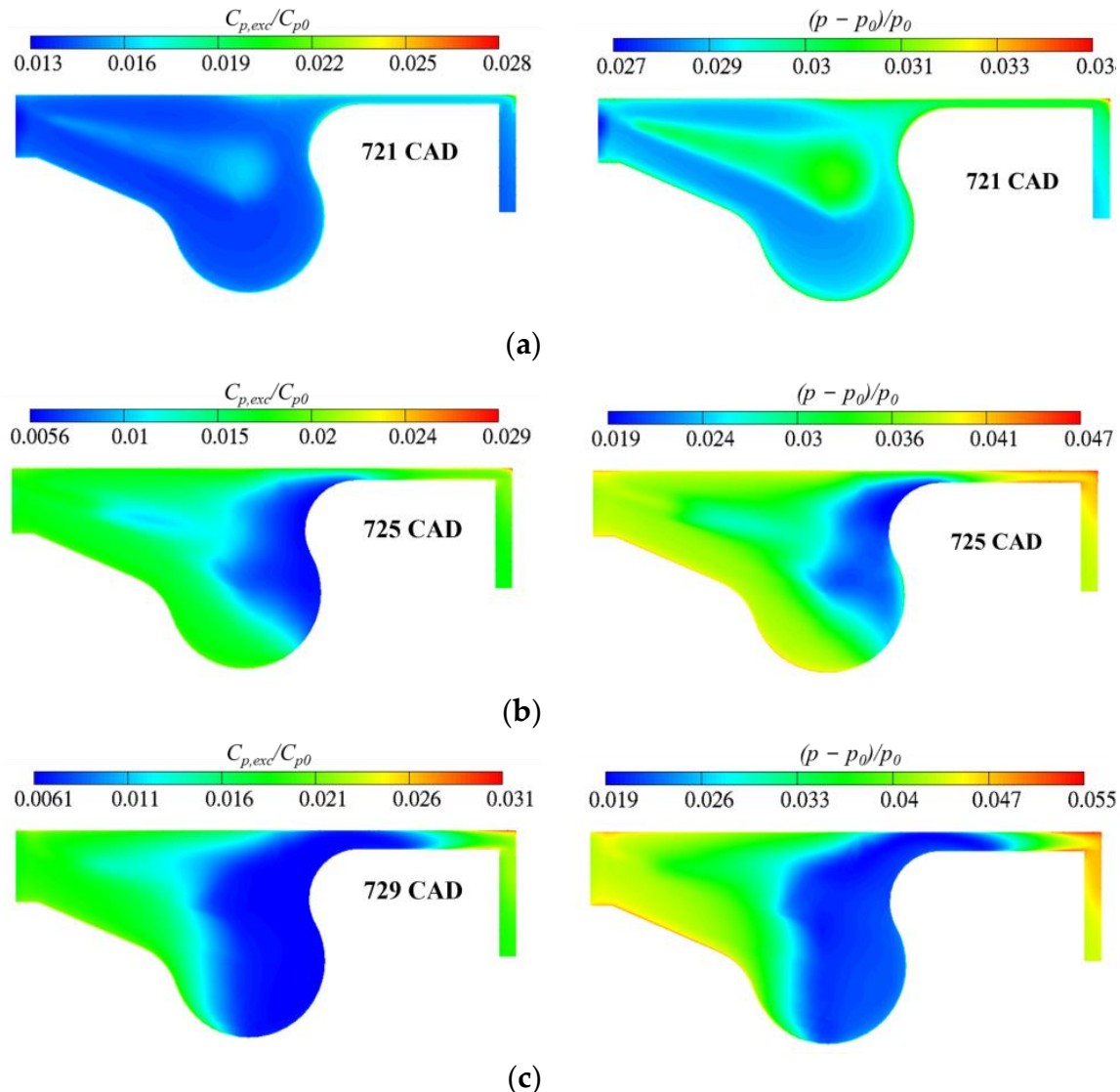

**Figure 20.** Predicted instantaneous distributions of the relative excess heat capacity $C_{p,\mathrm{exc}}/C_{p0}$ (left column) and the relative excess pressure $P_{\mathrm{exc}}/P_0$ (right column) in the combustion chamber at different times obtained in calculation III: (**a**) 721 CAD; (**b**) 725 CAD; and (**c**) 729 CAD.

Interestingly, studies of real gas effects on conventional diesel combustion with n-heptane as a fuel in [36] have shown somewhat different results. First, the Peng–Robinson real gas thermal and caloric EoS overpredicted both the TDC pressure and temperature in a single-cylinder diesel engine as compared to the ideal gas EoS, whereas our Figure 15a,b shows opposite trends. Second, the ignition timing has been found to be slightly advanced for the real gas case, whereas our Figure 15c shows a considerable delay in the ignition timing. Third, the heat release rate was found to be virtually not affected by the account for real gas properties, whereas our Figure 15c shows a considerable increase in the maximum heat release rate. Fourth, the real gas effects on NO and soot emissions have been found to be insignificant, whereas our Figure 18 shows a considerable effect on both NO and soot yields. On the one hand, the indicated qualitative differences could be caused by the chosen different operation modes of diesel engines as well as different computational settings. On the other hand, they could be caused by the poor accuracy of the Peng–Robinson EoS in the vicinity of the critical point of the fuel.

## 4. Conclusions

The computational studies reported in this paper show the influence of multistage fuel-oxidation chemistry, soot thermal radiation during droplet combustion, and real gas effects on the operation process of compression ignition engines. The use of the detailed reaction mechanism of iso-octane oxidation in the zero-dimensional simulation of the operation process in such an engine reveals the appearance of different combinations of cool, blue, and hot flames at different compression ratios and provides a kinetic interpretation to the phenomena, which essentially affect the heat release function and the engine operation process as a whole.

One-dimensional simulation of fuel droplet autoignition with regard to the detailed chemistry of multistage fuel oxidation and overall chemistry of soot formation shows that the effect of the thermal radiation of soot during autoignition and the subsequent combustion of small droplets under conditions of compression ignition engines turns out to be insignificant; the autoignition and combustion of droplets under such conditions can be approximately simulated without taking into account the thermal radiation of soot. Small (submillimeter size) droplets do not exhibit radiative flame quenching.

Comparative three-dimensional calculations of the operation process in a diesel engine were performed using thermal and caloric real gas and ideal gas equations of state. A significant effect of real gas properties on the engine indicator diagram and the yields of NO and soot was demonstrated. The relative excess heat capacity and relative excess pressure attained their maximum values (~2.5% and ~4%, respectively) near the cold walls of the combustion chamber and in the region of the liquid fuel spray; i.e., in the zones with high gas density. The account for real gas properties:

(1)  reduces the maximum pressure and mass-averaged temperature in the combustion chamber by about 7 bar (6%) and 150 K (9%), respectively;
(2)  increases the autoignition delay time by a 1.6 crank angle degree;
(3)  increases the maximum heat release rate by 20%; and
(4)  reduces the yields of NO and soot by a factor of 2 and 4, respectively.

Thus, the thermodynamic conditions in modern turbocharged diesel engines can go beyond the limits of the ideal gas approximation.

Future work will be focused on the three-dimensional simulation of the engine operation process with multistage fuel-oxidation chemistry and real gas effects taken into account. It is expected that the combined consideration of these effects will improve the predictability of calculations in terms of the heat release function and pollutant emissions.

**Author Contributions:** Conceptualization, V.Y.B. and S.M.F.; methodology, S.M.F.; validation, V.Y.B., S.M.F., V.S.I. and F.S.F.; formal analysis, V.Y.B. and S.M.F.; investigation, V.Y.B., S.M.F., V.S.I., F.S.F. and I.V.S.; resources, S.M.F.; data curation, V.S.I. and F.S.F.; writing—original draft preparation, V.Y.B. and S.M.F.; writing—review and editing, S.M.F.; supervision, S.M.F.; project administration, S.M.F.; funding acquisition, S.M.F. All authors have read and agreed to the published version of the manuscript.

**Funding:** This study was conducted within the scientific program of the National Center for Physics and Mathematics, section #2 "Mathematical Modeling on Zetta-scale and Exa-scale Supercomputers. Stage 2023–2025" and was partly funded by a subsidy given to the Semenov Federal Research Center for Chemical Physics of the Russian Academy of Sciences to implement the state assignment with registration number 122040500073-4 and by a subsidy given to the Federal State Institution "Scientific Research Institute for System Analysis of the Russian Academy of Sciences" to implement the state assignment on the topic No. FNEF-2022-0005 (Registration No. 1021060708369-1-1.2.1).

**Institutional Review Board Statement:** Not applicable.

**Informed Consent Statement:** Not applicable.

**Data Availability Statement:** Data will be available on request.

**Conflicts of Interest:** The authors declare no conflict of interest.

## Abbreviations

| | |
|---|---|
| 0D | Zero-dimensional |
| 1D | One-dimensional |
| 3D | Three-dimensional |
| CAD | Crank angle degree |
| CFD | Computational fluid dynamics |
| CIE | Compression ignition engines |
| CV | Compensation volume |
| DKM | Detailed kinetic mechanism |
| EoS | Equation of state |
| OM | Overall mechanism |
| RANS | Reynolds-averaged Navier–Stokes |
| SIMPLE | Semi-implicit method for pressure linked equations |
| TDC | Top dead center |
| TVD | Total variation diminishing |

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
