# Peer review of "The Effects of Multistage Fuel-Oxidation Chemistry, Soot Radiation, and Real Gas Properties on the Operation Process of Compression Ignition Engines"

_2673-4117, doi:10.3390/eng4040153_

Round 1

Reviewer 1 Report

This paper " The effects of multistage fuel-oxidation chemistry, soot radiation, and real gas properties on the operation process of compression ignition engines" has investigated the influence of multistage fuel-oxidation chemistry, thermal radiation of soot during combustion of a small (submillimeter size) fuel droplet, and real gas effects on the operation process of compression ignition engines. However, the manuscript has some defect. Thus, I suggest that the paper be major revision. The following hints may help the authors:

Q1: There are a lot of abbreviations in the article, and the full names are not indicated. It is necessary for the author to add a nomenclature at the beginning or end of the article. Avoid using abbreviations in the Title, Abstract and Conclusions, if possible.

Q2: Could you rewrite the abstract to be more focused on the main goal and the key obtained findings?

Q3: In the introduction, the advantage of multistage fuel-oxidation chemistry should be introduced, which has demonstrated the necessity of the study. Otherwise, the innovation of the paper and the necessity of research will not be perfectly displayed.

Q4: In paper, please summarize the main contribution of each references paper in separate sentences, such as [3-5, 17-19, 29-31]. The reference style should be checked again according to the journal standard.

Q5: In general,the conclusion is not well organized. The results should be further elaborated to show how they could be used for the real applications.

Q6: More details about the calculation of uncertainty analysis should be specified. What's the cause of the uncertainty analysis for performance parameters?

Q7: If possible, the author should consider that the fuel calorific value is equal for each cycle when the diesel engine fuels with different fuels.

Q8: Grid independence should be promoted.

Q9: In the simulation process, how did the author consider the fuel injection rate of diesel engines? How did the author obtain the fuel injection rate in the simulation model?

Q10:In the results and discussion section, the author is encouraged to provide more in-depth discussions about each figure, and not just an interpretation of what is shown in the Figures. In addition, the result trends should be compared with the existing literature; this is to ensure that they offer expected outcomes.

Moderate editing of English language required

Author Response

We are grateful to the reviewer for valuable comments. We have made our best to follow all the comments. All changes in the revised manuscript are marked in yellow.

This paper " The effects of multistage fuel-oxidation chemistry, soot radiation, and real gas properties on the operation process of compression ignition engines" has investigated the influence of multistage fuel-oxidation chemistry, thermal radiation of soot during combustion of a small (submillimeter size) fuel droplet, and real gas effects on the operation process of compression ignition engines. However, the manuscript has some defect. Thus, I suggest that the paper be major revision. The following hints may help the authors:

Q1: There are a lot of abbreviations in the article, and the full names are not indicated. It is necessary for the author to add a nomenclature at the beginning or end of the article. Avoid using abbreviations in the Title, Abstract and Conclusions, if possible.

We have added a table of abbreviations at the end of the manuscript. The abbreviations in the Title, Abstract and Conclusions are avoided.

Q2: Could you rewrite the abstract to be more focused on the main goal and the key obtained findings?

We have rewritten the abstract following the reviewer’s recommendations.

Q3: In the introduction, the advantage of multistage fuel-oxidation chemistry should be introduced, which has demonstrated the necessity of the study. Otherwise, the innovation of the paper and the necessity of research will not be perfectly displayed.

To address this comment, we have extended one sentence in the Introduction, namely:

“Moreover, such models and mechanisms do not usually include low-temperature chemical transformations in the combustible mixture, which can have a noticeable effect on the rate of energy release and engine operation process as a whole.”

Q4: In paper, please summarize the main contribution of each references paper in separate sentences, such as [3-5, 17-19, 29-31]. The reference style should be checked again according to the journal standard.

We have extended the info on the contributions of each indicated reference.

 Q5: In general,the conclusion is not well organized. The results should be further elaborated to show how they could be used for the real applications.

We have modified the Conclusions to be somewhat more specific and added the sentence on the future work:

“Future work will be focused on the three-dimensional simulation of the engine operation process with multistage fuel-oxidation chemistry and real gas effects taken into account.”

Q6: More details about the calculation of uncertainty analysis should be specified. What's the cause of the uncertainty analysis for performance parameters?

To address this comment, we have added several sentences to section 2. Materials and Methods and highlighted the existing sentences related to model validation:

In relation to Figure 2:

“Comparison of the calculation results with the experimental data is seen to yield their satisfactory agreement as the uncertainty in the ignition delay data can exceed 100% [44,45].”

In relation to Figures 3-5:

“In general, the agreement between calculations and measurements is encouraging keeping in mind that the accuracy of optical measurements of droplet diameter is moderate while the accuracy of ignition delay measurements can attain 100% [57].”

In relation to Table 1:

“It turns out that the error in calculating the pressure according to Eq. (24) does not normally exceed tenths of a percent even in the vicinity of the critical point.”

In relation to Figure 6:

“As seen, the OM of Table 2 provides satisfactory qualitative and quantitative agreement with the results predicted by the DKM [69].”

Q7: If possible, the author should consider that the fuel calorific value is equal for each cycle when the diesel engine fuels with different fuels.

In this study, we apply the common practice of using diesel oil as a physical surrogate and n-heptane as a chemical surrogate [R1]. This means that the fuel spray behaves (penetrates, breaks up, evaporates, etc.) like the spray of diesel oil, whereas the chemical kinetics of fuel vapor oxidation is represented by n-heptane reaction mechanism. Although such a coupling is still an open question, the performance of the mechanism in the simulation is usually very satisfactory both in terms of the predicted pressure and heat release rate curves. As stated in [R1], the prediction of ignition timing and combustion phasing might be further improved if a real diesel mechanism would have been used in the simulation.

[R1] Feng Tao, Yi Liu, Bret A. RempelEwert, David E. Foster, Rolf D. Reitz, Dae Choi, Paul C. Miles, Modeling the Effects of EGR and Injection Pressure on Soot Formation in a High-Speed Direct-Injection (HSDI) Diesel Engine Using a Multi-Step Phenomenological Soot Model, SAE Paper 2005-01-0121 (2005).

To follow this comment, we have added a new reference [R1]=[77] to the manuscript and renumbered all subsequent references.

Q8: Grid independence should be promoted.

This issue has been already addressed in the original manuscript at the end of subsection 2.5 (we have highlighted the corresponding statement):

“Preliminary mesh sensitivity studies proved this spatial resolution to be sufficient.”

Q9: In the simulation process, how did the author consider the fuel injection rate of diesel engines? How did the author obtain the fuel injection rate in the simulation model?

The AVL FIRE code [70] offers a standard procedure of liquid fuel injection. The user sets the injection timing (start of injection and end of injection), injected mass, and injection angle. These parameters are presented in Table 3. The function of fuel injection rate vs. time depends on the type of fuel injector and is provided by the code.

Q10:In the results and discussion section, the author is encouraged to provide more in-depth discussions about each figure, and not just an interpretation of what is shown in the Figures. In addition, the result trends should be compared with the existing literature; this is to ensure that they offer expected outcomes.

To follow this comment, we have extended discussions where appropriate.

Reviewer 2 Report

The submitted work entitled The effects of multistage fuel-oxidation chemistry, soot radiation, and real gas properties on the operation process of compression ignition engines contains theoretical considerations on the subject matter. The work is based on assumptions derived from many previous studies by the authors. It is not possible to isolate what innovations the authors were carrying out in their work. The structure of the work is also disturbed. Therefore, the manuscript needs to be corrected and clarified based on the following comments:

1. The work contains theoretical considerations and computer simulations based on assumptions from the authors' work. The work contains excessive self-citations in the form of references to assumptions from previous works, often difficult to check due to limited access. In the methodological part, the authors in many places refer to the works of other authors, works related thematically but published quite a long time ago. It is difficult to catch the novelty of the work, as well as the differences between the manuscript and other works by the authors. This should be clearly described in the manuscript.

2. The literature review is based on numerous thematically consistent items, unfortunately coming from a limited group of authors, and above all in the form of references to the authors' own works.

3. An abstract that is too long and too general, does not reflect the entire content of the paper and points to a review rather than a research article.

4. Introduction is a monographic chapter with too much discussion of basic and generally known issues. There you can find several repetitions and theoretical considerations that correspond to the assumptions of the methodological part. The literature review was not separated either.

5. The methodology is based primarily on references to the work of other authors. Many assumptions and mental shortcuts were made by referring to previous works, which makes it impossible to isolate the authors' contribution to the work.

6. There is no indication of the practical, engineering use of the obtained simulation and interpretation results.

7. Extensive literature review, includes numerous items that have limited access and date back 10 years or more.

Author Response

We are grateful to the reviewer for valuable comments. We have made our best to follow all the comments. All changes in the revised manuscript are marked in green.

The submitted work entitled The effects of multistage fuel-oxidation chemistry, soot radiation, and real gas properties on the operation process of compression ignition engines contains theoretical considerations on the subject matter. The work is based on assumptions derived from many previous studies by the authors. It is not possible to isolate what innovations the authors were carrying out in their work. The structure of the work is also disturbed. Therefore, the manuscript needs to be corrected and clarified based on the following comments:

  1. The work contains theoretical considerations and computer simulations based on assumptions from the authors' work. The work contains excessive self-citations in the form of references to assumptions from previous works, often difficult to check due to limited access. In the methodological part, the authors in many places refer to the works of other authors, works related thematically but published quite a long time ago. It is difficult to catch the novelty of the work, as well as the differences between the manuscript and other works by the authors. This should be clearly described in the manuscript.

To address this comment, we have added a statement on the novelty of this work at the end of Introduction section:

“These objectives and the obtained results are the novel and distinctive features of this paper.”

  1. The literature review is based on numerous thematically consistent items, unfortunately coming from a limited group of authors, and above all in the form of references to the authors' own works.

Our literature review is aimed at providing the status of the subjects we study. We could readily duplicate the number of references, but restricted ourselves by the pioneering and most outstanding publications. Our own works are cited because we use our own reaction mechanisms, droplet models, and real gas equations of state for solving the posed problems.

  1. An abstract that is too long and too general, does not reflect the entire content of the paper and points to a review rather than a research article.

We have rewritten the Abstract.

  1. Introduction is a monographic chapter with too much discussion of basic and generally known issues. There you can find several repetitions and theoretical considerations that correspond to the assumptions of the methodological part. The literature review was not separated either.

To address this comment, we have restructured the Introduction by introducing three subsections focusing on the particular subject of the present study.

  1. The methodology is based primarily on references to the work of other authors. Many assumptions and mental shortcuts were made by referring to previous works, which makes it impossible to isolate the authors' contribution to the work.

As a matter of fact, the corresponding models and solution methodologies are based on OUR OWN WORK. Therefore, we briefly introduce the models in sections 2.1–2.3 and solution methodologies in sections 2.4–2.6 of this manuscript with the corresponding references to our previous publications. The problems solved in this work are different from those solved in our previous works. Thus, for the first time we have applied our reaction mechanism of iso-octane oxidation to compression ignition engine and provided the kinetic interpretation to the classical diagram of Figure 1. For the first time we applied our model of droplet ignition and combustion to engine conditions and proved the negligible effect of soot thermal radiation on droplet evolution. Finally, for the first time we have applied our real gas equation of state to engine conditions and demonstrated a significant effect of real gas properties on the engine operation process. The next step will be the combination of all these models in engine simulation (this is indicated at the end of Conclusions).

To address this comment, we have added the following paragraph at the end of Introduction:

“The paper is organized as follows. In Section 2, we briefly introduce our own DKM of hydrocarbon fuel oxidation (Subsection 2.1), our own model of droplet autoignition and combustion (Subsection 2.1), and our own real gas EoS (Subsection 2.3) with the examples of their validation. Thereafter we briefly describe the procedures of the numerical solution of the new target problems, namely, the manifestation of multi-stage autoignition in the CIE (Subsection 2.4), the manifestation of soot thermal radiation during droplet combustion in CIE (Subsection 2.5), and the manifestation of real gas effects in CIE (Subsection 2.6). In Section 3, we show the results of calculations and discuss the corresponding implications. Section 4 summarizes the results of the study and discuss the directions of future work.”

  1. There is no indication of the practical, engineering use of the obtained simulation and interpretation results.

To follow this comment, we have added a sentence at the end of Conclusions:

“It is expected that the combined consideration of these effects will improve the predictability of calculations in terms of the heat release function and pollutant emissions.”

  1. Extensive literature review, includes numerous items that have limited access and date back 10 years or more.

All references are available in the net. We could provide the reviewer with any reference from the list upon request.

Reviewer 3 Report

Review

Manuscript Number: eng-2599537

Full Title: The effects of multistage fuel-oxidation chemistry, soot radiation, and real gas properties on the operation process of compression ignition engines

Comments and Suggestions for Authors

l  General comments:

- Bibliography is enough, actual and relevant;

- English needs revision;

- Subject is interesting and actual.

l  Detailled comments:

(1)   The operation process of compression ignition engines was studied by simulation in this work. Overall, the research is valuable for the effects of multistage fuel-oxidation chemistry, soot radiation, and real gas properties.

(2)   Subject is actual and very interesting. English must be improved. Some sentences are confuse and some spell check is required. English language should be carefully checked and carefully check paper for language typos.

(3)   Please pay more effort to improve the paper so that it addresses the following points sufficiently.

²  Added the values to previous researches

²  Added the model experiment validation method

²  Explain the source of the boundaries data and the simulation method

(4)   Abstract just briefly describe the importance of the research and validation method, and key results obtained could be the added values to the knowledge in the subject. If these points are correct please reorganise and improve the contents to be centralised on this points.

(5)   In the parts of abstract and conclusion, it is necessary to add in-depth scientific theoretical analysis, rather than simply describing the results. At the end of the introduction section, there is no information as to how an article is organized. What is included in the following sections. Conclusion section is missing some perspective related to the future research work.

(6)   Introduction: It is suggested to add a summary description of the research status of the real gas properties. Please cite more relevant studies.

(7)   In Section 2.6 Solution proceure of the three-dimensional problem, the following questions need to be addressed.

²  added the experimental verification scheme

²  simplified geometric modeling

²  adaptive analysis of mesh scale

(8)   In 3.3 Real gas effects, the instantaneous temperature distributions in the combustion chamber need to be detailed for Fig. 15, Fig. 16. The scales in the cloud image should be consistent.

(9)   This paper need been minor revision and reviewed before publication.

That’s all. Thank you.

Minor editing of English language required.

English must be improved. Some sentences are confuse and some spell check is required. English language should be carefully checked and carefully check paper for language typos.

Author Response

We are grateful to the reviewer for valuable comments. We have made our best to follow all the comments. All changes in the revised manuscript are marked in blue.

l  General comments:

- Bibliography is enough, actual and relevant;

- English needs revision;

We have double checked English grammar and spelling. Also, we asked our native English-speaking college to check the text.

- Subject is interesting and actual.

l  Detailled comments:

  • The operation process of compression ignition engines was studied by simulation in this work. Overall, the research is valuable for the effects of multistage fuel-oxidation chemistry, soot radiation, and real gas properties.
  • Subject is actual and very interesting. English must be improved. Some sentences are confuse and some spell check is required. English language should be carefully checked and carefully check paper for language typos.

We have double checked English grammar and spelling. Also, we asked our native English-speaking college to check the text.

(3)   Please pay more effort to improve the paper so that it addresses the following points sufficiently.

²  Added the values to previous researches

Actually, at the end of Section 3.3 we compare our results with the results reported by other researchers:

“Interestingly, the studies of real gas effects on the conventional diesel combustion with n-heptane as a fuel in [4] show somewhat different results. First, the Peng–Robinson real gas thermal and caloric EoS overpredict both the TDC pressure and temperature in a single cylinder diesel engine as compared to the ideal gas EoS, whereas our Figures 14a and 14b show opposite trends. Second, the ignition timing is found to be slightly advanced for the real gas case, whereas our Figure 14c shows a considerable delay in the ignition timing. Third, the heat release rate is found to be virtually not affected by the account for real gas properties, whereas our Figure 14c shows a considerable increase in the maximum heat release rate. Fourth, the real gas effects on NO and soot emissions are found to be insignificant, whereas our Figure 17 shows a considerable effect on both NO and soot yields. On the one hand, the indicated qualitative differences could be caused by the chosen different operation modes of diesel engines as well as different computational settings. On the other hand, they could be caused by poor accuracy of the Peng–Robinson EoS in the vicinity of the critical point of fuel.”

²  Added the model experiment validation method

To address this comment, we have included a new Figure 8 to compare the results of calculations with measurements.

²  Explain the source of the boundaries data and the simulation method

To address this comment, we have added a footnote to Table 3: “The data are provided by AVL LIST GmbH”

Abstract just briefly describe the importance of the research and validation method, and key results obtained could be the added values to the knowledge in the subject. If these points are correct please reorganise and improve the contents to be centralised on this points.

Following this comment, we have extended the abstract by introducing the most important findings.

(5)   In the parts of abstract and conclusion, it is necessary to add in-depth scientific theoretical analysis, rather than simply describing the results. At the end of the introduction section, there is no information as to how an article is organized. What is included in the following sections. Conclusion section is missing some perspective related to the future research work.

As a matter of fact, this material (how the article is organized and future work) was added to the manuscript at the first review round to follow the comments of the Reviewer #2:

Intoduction:

“The paper is organized as follows. In Section 2, we briefly introduce our own DKM of hydrocarbon fuel oxidation (Subsection 2.1), our own model of droplet autoignition and combustion (Subsection 2.1), and our own real gas EoS (Subsection 2.3) with the examples of their validation. Thereafter we briefly describe the procedures of the numerical solution of the new target problems, namely, the manifestation of multi-stage autoignition in the CIE (Subsection 2.4), the manifestation of soot thermal radiation during droplet combustion in CIE (Subsection 2.5), and the manifestation of real gas effects in CIE (Subsection 2.6). In Section 3, we show the results of calculations and discuss the corresponding implications. Section 4 summarizes the results of the study and discuss the directions of future work.”

Conclusions:

“Future work will be focused on the three-dimensional simulation of the engine operation process with multistage fuel-oxidation chemistry and real gas effects taken into account. It is expected that the combined consideration of these effects will improve the predictability of calculations in terms of the heat release function and pollutant emissions.”

(6)   Introduction: It is suggested to add a summary description of the research status of the real gas properties. Please cite more relevant studies.

We have added some more references dealing with real-gas effects to the Introduction.

(7)   In Section 2.6 Solution proceure of the three-dimensional problem, the following questions need to be addressed.

²  added the experimental verification scheme

We have added a new Figure 8 and the following text to demonstrate the predicting ability of the 3D approach:

“Figure 8 demonstrates the predictive capabilities of the model. It compares the measured pressure in the diesel engine under consideration with the pressure histories calculated using the base and fine meshes. In the calculations, the ideal-gas thermal and caloric EoS are used. To fit the measured pressure curve, the empirical constants of the WAVE model were slightly adjusted (within the recommended range) as compared to the default values (= 0.61 and  = 30), but were the same for both meshes. Despite the use of the simplified engine model, the calculated pressure histories are seen to agree satisfactorily with each other and with the measurements.”

In view of adding the new Figure 8, all subsequent Figures are reordered.

²  simplified geometric modeling

In Section 2.6, we discuss the simplifications adopted in the 3D model:

“It is assumed that all cylinders of the engine operate in the same way, so only one engine cylinder is considered in the calculations. If we assume that the combustion process is axisymmetric, we can proceed to the consideration of a segment model, which is 1/8 of the combustion chamber in accordance with the number of nozzles in the injector, which greatly simplifies the geometry of the computational domain. However, such a simplification requires the account for the volume of technological recesses in the geometry of the combustion chamber associated with valves, etc. This is usually made by providing the so-called compensation volume (CV) in the geometry of the computational domain. The CV is an additional volume equal to the unaccounted volume of recesses in the piston and in the head cylinder block [71]. In this case, the CV is placed along the cylinder wall. Figure 7 shows a base (Figure 7a) and fine (Figure 7b) computational meshes used in the calculations. The average cell size in the base and fine meshes is 0.5 and 0.25 mm, respectively. The computational mesh is movable.”

To further address this issue, we have added another sentence at the end of section 2.6:

“Despite the use of the simplified engine model, the calculated pressure histories are seen to agree satisfactorily with each other and with the measurements.”

²  adaptive analysis of mesh scale

To follow this comment, we have extended Figure 7 to include both base and fine computational meshes, and showed two calculated pressure curves in Figure 8 obtained using these meshes. Also, we added the sentence:

“This comparison justifies the use of the base computational mesh for simulating the engine operation process.”

Also, we added the following important note to Section 3.3:

“All calculations are made using the base mesh of Figure 7a under completely identical model settings as well as initial and boundary conditions.”

(8)   In 3.3 Real gas effects, the instantaneous temperature distributions in the combustion chamber need to be detailed for Fig. 15, Fig. 16. The scales in the cloud image should be consistent.

We have modified new Figures 16 and 17 by adopting the same temperature scale.

(9)   This paper need been minor revision and reviewed before publication.

Done.

That’s all. Thank you.

Round 2

Reviewer 1 Report

The authors have carried out a thorough and careful revision and the revised manuscript improved a lot in terms of technical quality and language. Therefore, I would recommend it for publication in the Journal.

The authors have carried out a thorough and careful revision and the revised manuscript improved a lot in terms of technical quality and language. Therefore, I would recommend it for publication in the Journal.

Author Response

We are grateful to the reviewer for valuable comments.

Reviewer 2 Report

Comments from the reviews in the first round were only minimally taken into account or were not taken into account at all. The work contains excessive self-citations, most of the references are works older than 10 years.

Author Response

We are grateful to the reviewer for valuable comments. We have made our best to follow all the comments. All changes in the revised manuscript for both rounds of the review are marked in green.

Round 2

Comments from the reviews in the first round were only minimally taken into account or were not taken into account at all. The work contains excessive self-citations, most of the references are works older than 10 years.

We have made our best to address all comments of the first round.

To address the first comment of the second round, we have found possible to delete several references to our own papers (former Refs. 17, 18, 32, 35, 66) and correspondingly change some sentences and renumber other references. Now, the rate of self-citation is 12%, which seems acceptable.

To follow the second comment that most of references are older than 10 years, we have added 4 more most recent references on the real-gas effects. Now, “old” references are only 47 of 78, i.e., 60%. As we mentioned earlier, our literature review is aimed at providing the genesis and current status of the problems we study. We restricted ourselves by the pioneering and most outstanding classical and up-to-date publications. Our own works are cited because we use our own reaction mechanisms, droplet models, and real gas equations of state and must indicate the corresponding literature sources.

Round 3

Reviewer 2 Report

Not all comments have been fully addressed in the manuscript. Only 4 literature items were changed in the work. Citations still cover only a narrow group of authors. And there are definitely too many self-citations.

Author Response

We are grateful to the reviewer for the comments. To follow the comments, we have thoroughly rearranged the Introduction section, deleted two more self-citations, added 7 more most recent references with the corresponding text, reordered equations and references. All changes are marked in green.